# Genomic insights into antibiotic resistance and mobilome of lactic acid bacteria and bifidobacteria

Vita Rozman[1] , Petra Mohar Lorbeg[1], Primož Treven[1], Tomaž Accetto[2], Sandra Janežič[3,4], Maja Rupnik[3,4], Bojana Bogovič Matijašić[1]

Lactic acid bacteria (LAB) and *Bifidobacterium* sp. (bifidobacteria) can carry antimicrobial resistance genes (ARGs), yet data on resistance mechanisms in these bacteria are limited. The aim of our study was to identify the underlying genetic mechanisms of phenotypic resistance in 103 LAB and bifidobacteria using whole-genome sequencing. Sequencing data not only confirmed the presence of 36 acquired ARGs in genomes of 18 strains, but also revealed wide dissemination of intrinsic ARGs. The presence of acquired ARGs on known and novel mobile genetic elements raises the possibility of their horizontal spread. In addition, our data suggest that mutations may be a common mechanism of resistance. Several novel candidate resistance mechanisms were uncovered, providing a basis for further in vitro studies. Overall, 1,314 minimum inhibitory concentrations matched with genotypes in 92.4% of the cases; however, prediction of phenotype based on genotypic data was only partially efficient, especially with respect to aminoglycosides and chloramphenicol. Our study sheds light on resistance mechanisms and their transferability potential in LAB and bifidobacteria, which will be useful for risk assessment analysis.

## Introduction

Antimicrobial (antibiotic) resistance of foodborne pathogenic bacteria is an important food safety problem (1). Commensal bacteria, including lactic acid bacteria (LAB) and *Bifidobacterium* sp. (bifidobacteria), have recently been recognised as a reservoir of resistance genes (ARGs) (2, 3, 4). They are introduced into the agro-food chain as starter and probiotic cultures, protective cultures, and feed additives. Because they come into contact with bacteria residing in gut—a hotspot of microbial horizontal gene transfer (5)—they pose a risk for transmission of ARGs. In scope of the Qualified Presumption of Safety status, such strains

must be free of acquired ARGs (6). It was not until 2018 that the guidelines for the characterisation of microorganisms used as feed additives or as production organisms (7) included a requirement for strain characterisation based on whole-genome sequences (WGS). Since then, several studies have analysed resistance genes in LAB and bifidobacteria based on WGS (2, 3, 4, 8, 9), but still only a handful of studies have focused on strains intentionally added to the agro-food chain (10, 11). In addition, these studies often lack data on intrinsic and mutational resistance and transfer capability of ARGs through mobile genetic elements (MGEs). An in-depth understanding of the resistance mechanisms and their potential for transferability is essential to ensure the safety of dietary supplements (probiotics), feed additives, and products manufactured with starter or protective cultures.

Given that most antimicrobials are natural compounds, innate resistance mechanisms have evolved over time. Such natural (intrinsic) antimicrobial resistance is inherent to the species and presents a minimal potential for horizontal spread (12). On the contrary, resistance can be acquired either by a novel genetic mutation of chromosomal genes or by added resistance genes by means of horizontal gene transfer. Resistance acquired through added gene(s) is considered to have a high potential for horizontal dissemination (12). Acquired resistance in *Enterococcus* sp. (enterococci) is widespread and considerably well described, as some strains are important nosocomial pathogens (3, 13). On the contrary, data on resistance, particularly on intrinsic ARGs and mutations, are not as comprehensive in other genera of LAB and bifidobacteria.

In this context, the main goals of our study were not only to determine phenotypic susceptibility of LAB and bifidobacteria from different sources to antimicrobials but also to identify the potential underlying mechanisms of acquired and intrinsic resistance. In addition, we aimed to discover known and novel MGEs using whole-genome sequencing and comparative genomics. To achieve these goals, we analysed 103 strains, mainly commercial cultures but also isolates from human milk and from fermented products of which the genomes of 75 strains were sequenced in-house.

---

[1]University of Ljubljana, Biotechnical Faculty, Department of Animal Science, Institute of Dairy Science and Probiotics, Domžale, Slovenia  [2]University of Ljubljana, Biotechnical Faculty, Department of Microbiology, Chair of Microbial Diversity, Microbiomics and Biotechnology, Ljubljana, Slovenia  [3]National Laboratory of Health, Environment and Food, Maribor, Slovenia  [4]University of Maribor, Faculty of Medicine, Maribor, Slovenia

Correspondence: vita.rozman@bf.uni-lj.si

# Results

## Antimicrobial resistance phenotypes

LAB and bifidobacteria can carry mobile ARGs, and when ingested, they can facilitate the transfer of these genes to the resident microbiota in the gut and thus to potential pathogens. Commercial strains are required to be free of acquired (mobile) ARGs (6), but data on the genetic basis for phenotypic resistance in these bacteria are limited. The main objective of our study was to identify the potential underlying mechanisms of acquired and intrinsic resistance in LAB and bifidobacteria using comparative genomics.

The minimum inhibitory concentrations (MICs) of up to 27 antimicrobials were tested using the broth microdilution method for 103 LAB and bifidobacteria (Fig 1). We observed that resistance to kanamycin and resistance to chloramphenicol were the most common clinically relevant phenotypes (Fig 1). In contrast, the lower prevalence of resistance was seen with gentamicin, erythromycin, and ampicillin, whereas atypical vancomycin resistance (7) was not detected (Fig 1). Multidrug resistance frequently occurred in *Enterococcus* sp., *Levilactobacillus brevis*, *Lacticaseibacillus rhamnosus*, and *Pediococcus* sp. Surprisingly, three strains showed resistance to five groups of clinically important antimicrobials (Fig 1).

## Whole-genome sequence analysis

### Acquired ARGs

Genomic data (n = 103) were mined for the presence of ARGs whose intrinsic or acquired nature was determined by MGEs and pan-genome analyses. Based on the selection criteria, a total of 36 acquired ARGs corresponding to 18 diverse reference ARGs were found in 18 strains (Fig 2 and Table S1). Most of these ARGs (n = 33) were expressed in the resistant phenotype. Collectively, these genes conferred resistance to a broad array of antimicrobial classes (Table S1).

Analysis revealed that the tetracycline resistance gene *tetW* was most frequently detected, particularly in *Bifidobacterium* (*B.*) *animalis* subsp. *lactis* strains and in a probiotic *Limosilactobacillus reuteri* (Fig 2). Phenotypic tetracycline, but not tigecycline, resistance was less frequently conferred by the *tet*(L), *tet*(M), *tet*(O), *tet*(S), or *tet*(U) genes that were found in isolates from the natural microbiota of fermented products (referred to as non-starter strains) *Enterococcus* (*E.*) *faecalis*, *Lactococcus* (*L.*) *lactis*, and *E. italicus*, and in a probiotic strain of *B. breve* (Fig 2). We found that streptomycin resistance in *E. faecalis* and *L. lactis* (Fig 2) was associated with *ANT(6)-Ia* that in enterococci appeared to be linked to *SAT-4* and *APH(3′)-IIIa*, the genes responsible for resistance to streptothricin, and kanamycin and neomycin, respectively. In addition, a bifunctional *AAC(6′)-Ie-APH(2″)-Ia* that reflected in atypical gentamicin, kanamycin, and neomycin MICs was found in *E. faecalis* (Fig 2).

The MLS$_B$ phenotype in *E. faecalis* and in a probiotic strain *B. longum* (Fig 2) was encoded by *erm*(B) and *erm*(49), respectively. Markedly, we found a known mutation upstream of *erm*(B) in all three enterococcal strains (TAAA duplication between −124 and −127

resulting in a premature stop codon of the leader peptide first reported by Oh et al (14)), which also facilitated resistance to the 16-membered macrolide tylosin, presumably because of the gene overexpression. Chloramphenicol resistance and trimethoprim resistance in *E. faecalis* (Fig 2) were attributed to the *cat* and *dfrG* gene, respectively, whereas the low level of ampicillin and penicillin resistance in *Carnobacterium* (*C.*) *divergens* may be due to the expression of *CAD-1 β*-lactamase.

### Intrinsic and candidate ARGs

Most of the ARGs were recognised as intrinsic (140 ARGs in 37 strains, of which 20 were diverse based on gene homology) (Fig 2 and Table S1). Consistent with the intrinsic aminoglycoside resistance phenotype, *E. faecium* strains contained *AAC(6′)-Ii* and *efmM*, whereas *E. hirae* and *E. durans* possessed *AAC(6′)-Iid* and *AAC(6′)-Iih*, respectively. Interestingly, we identified homologs of EfmM with a conserved active site (C185, C235) in the vast majority of the LAB species studied (Fig 2). The observed high aminoglycoside MICs in *B. longum* and *B. breve* appear to be connected to aminoglycoside phosphotransferases, though homologs were also found in *B. animalis.* The *efmA* gene with a surprisingly diverse sequence was present in *E. faecium* strains.

A total of 331 candidate ARGs were discovered (Fig 2 and Table S1), representing 33 diverse genes in 92 strains (37 species). These genes confer resistance to various antibiotics (Table S1). Interestingly, among these genes *arr-4* in *L. lactis* IM145 had conserved amino acid residues His18, Tyr48, and Asp83, which are involved in rifampicin resistance (15). *E. malodoratus* IM1302 encompassed *fosXCC* with conserved amino acid residues in its active site (His7, His64, Glu110, Tyr100, and Arg119), which has been linked to resistance to fosfomycin in *Campylobacter* (16). Although most of these genes were presumably intrinsic, we also found acquired candidate ARGs (e.g., *arr-4, catB9, lnuA, ANT(6), mefA, vga(E)*). Their actual involvement in the resistance phenotype remains to be verified in vitro.

### Mutations associated with antimicrobial resistance

We provide comprehensive data on mutations in proteins previously reported to be involved in resistance (Table S2). The results suggest that mutations may be an important mechanism of resistance, particularly in bacteria intentionally introduced into the agro-food chain. We discovered known mutations already reported in the studied species, as well as novel mutations in the active (binding) sites of the target (or other) proteins not yet reported in the species or genera considered. Their role in resistance should be further elucidated in vitro.

Multiple sequence alignment of the S12 proteins revealed two substitutions, K43R/N/M and K88Q (*Mycobacterium tuberculosis* numbering), in commercial streptomycin-resistant strains (Fig 3A). Likewise, we discovered *rsmG* point mutations (I55A, G164V, and D67N, *M. tuberculosis* numbering; and G10E and R190H, *Streptomyces coelicolor* numbering, Table S2) involved in low-level streptomycin resistance. Although LAB are generally less susceptible to aminoglycosides, three strains (*Lactobacillus acidophilus* IM116, and *L. lactis* IM1456, IM1341) exhibited a hypersusceptible phenotype. Interestingly, these strains harboured single nucleotide polymorphisms (SNPs) in the F0F1

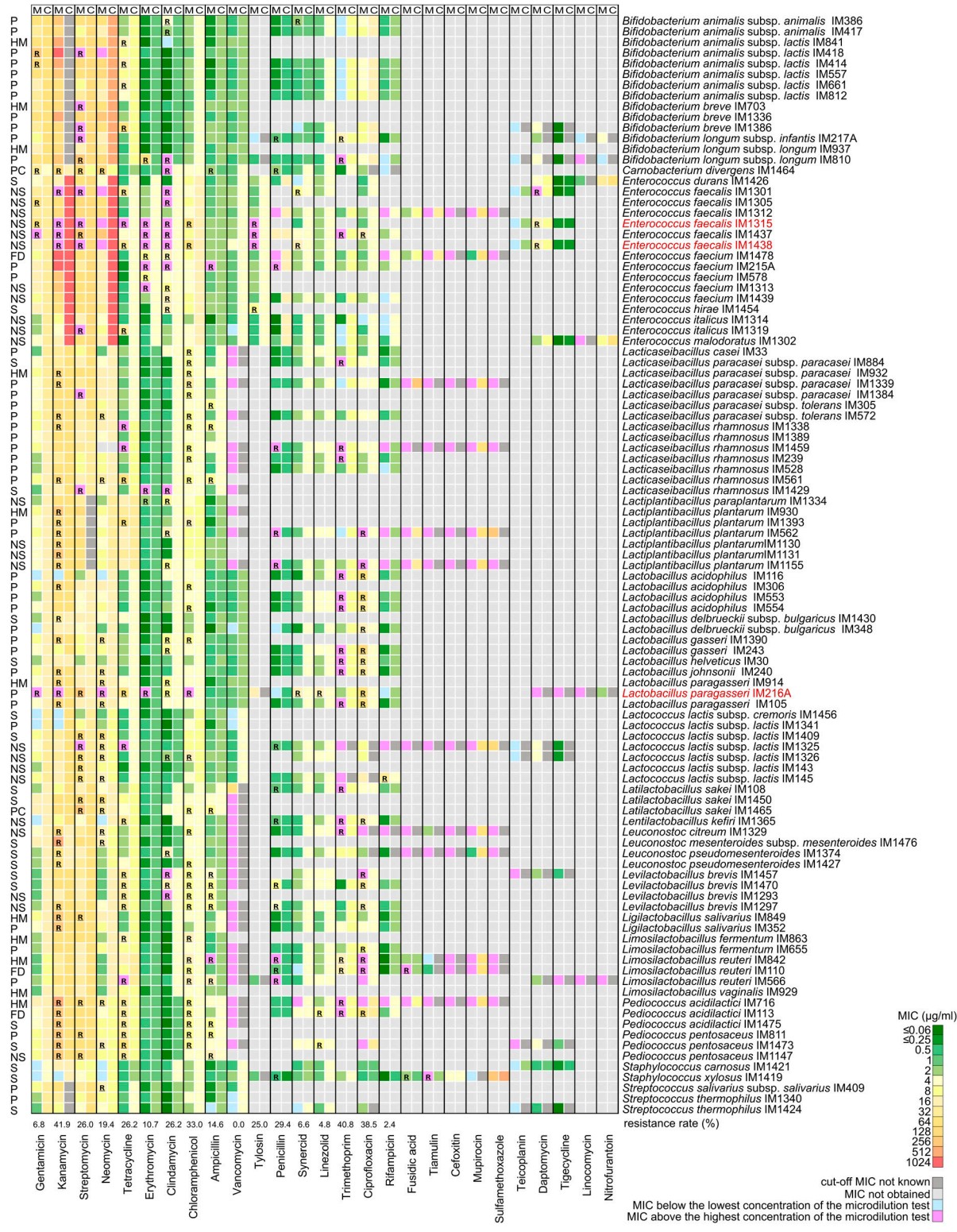

**Figure 1. Phenotypic resistance profiles of 103 lactic acid bacteria and bifidobacteria.**
The minimum inhibitory concentrations (MICs) determined by the microdilution tests and the cut-off MICs that define whether a strain is susceptible or resistant to a particular antibiotic are shown as a heatmap. The names of the strains resistant to five different classes of clinically important antimicrobials are highlighted in red. C, cut-off MICs; FD, feed additive; HM, isolate from human milk or colostrum; M, MICs determined by microdilution tests; NS, isolate of natural microbiota from fermented products (non-starter strain); P, probiotic strain; PC, protective culture; R, resistance; Synercid, quinupristin/dalfopristin; S, starter culture.

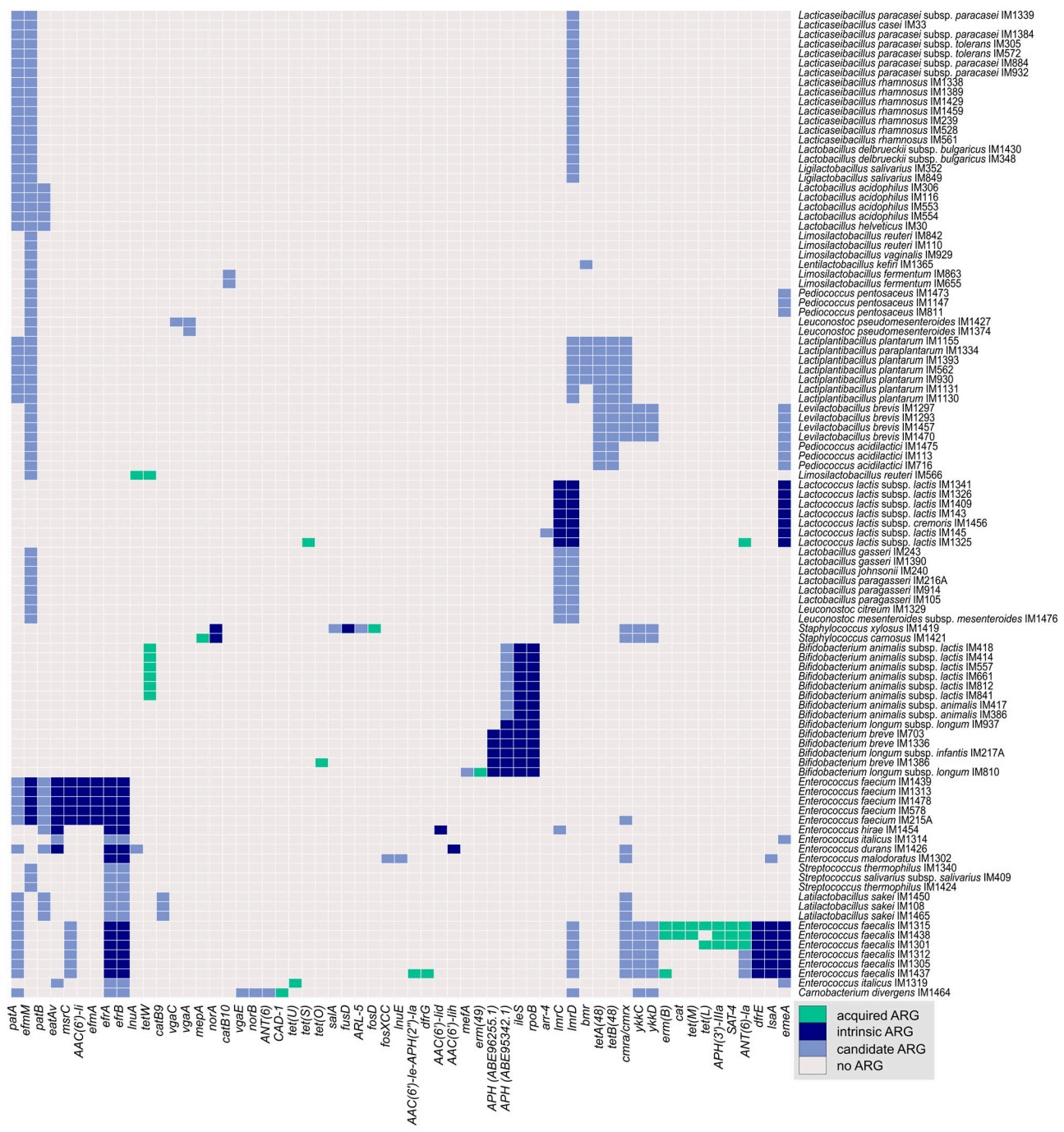

**Figure 2. Acquired, intrinsic, and candidate antimicrobial resistance genes (ARGs) found in 103 bacterial strains.**
A gene was annotated as an ARG based on the best BLAST hit with a sequence similarity threshold greater than 70%. The intrinsic and acquired nature of ARGs was determined with the aid of mobile genetic element prediction and pan-genome analyses. Candidate (homologous) ARGs were identified based on additional analyses of the hits with lower BLAST similarities (sequence similarity threshold between 40% and 70%).

ATPase genes (Table S2). F0F1 ATPase is reportedly involved in aminoglycoside transport into cells (17) that could be hampered by these mutations. The effects of these mutations on resistance have yet to be confirmed in vitro.

Several resistant probiotic strains had SNPs in *16S rRNA* (A1408G, C1054T, and A1197T, *E. coli* numbering) that presumably confer

resistance to aminoglycosides or tetracycline (Fig 3B and C). We also identified a SNP (A986T, *E. coli* numbering) near the primary tetracycline binding site in the representatives of LAB (Table S2) displaying high-end tetracycline MICs. Among four SNPs in *23S rRNA* (Fig 3B and C), G2057T, A2058G, and C2610T presumably encode resistance to MLS$_B$, whereas A2062T encodes resistance to tylosin,

**A.**

|  |  | 43 |  | MIC (µg/ml) |
|---|---|---|---|---|
| *M. tuberculosis* CDC1551 | CTRVYTTTP**K**KPNS | 47 | | S |
| *B. animalis* subsp. *animalis* IM417 | CTRVYTTTP**K**KPNS | 47 | | 8 |
| *B. animalis* subsp. *lactis* IM661 | CTRVYTTTP**K**KPNS | 47 | | 8 |
| *B. longum* subsp. *infantis* IM217A | CTRVYTTTP**N**KPNS | 47 | | **>1024** |
| *B. breve* IM1386 | CTRVYTTTP**R**KPNS | 47 | | **>256** |
| *B. breve* IM703 | CTRVYTTTP**R**KPNS | 47 | | **>256** |
| *B. longum* subsp. *longum* IM937 | CTRVYTTTP**K**KPNS | 47 | | 16 |
| *B. breve* IM1336 | CTRVYTTTP**K**KPNS | 47 | | 128 |
| *Lcb. paracasei* IM1339 | ATRVGTMTP**K**KPNS | 60 | | 32 |
| *Lcb. paracasei* IM1384 | ATRVGTMTP**R**KPNS | 60 | | **>256** |
| *Lcb. rhamnosus* IM1429 | ATRVGTMTP**M**KPNS | 60 | | **>256** |
| *Lcb. rhamnosus* IM1459 | ATRVGTMTP**K**KPNS | 60 | | 32 |
| | .*** * ** **** | | | |

**B.**

| | *16S rRNA* | | *23S rRNA* | | | | MIC (µg/ml) | | | | | | |
|---|---|---|---|---|---|---|---|---|---|---|---|---|---|
| | 1408 | 1197 | 2057 2062 | 2610 | GEN | KAN | NEO | TET | ERY | CHL | QDA | LIN |
| K–12 | CACAC | CAAGTC | GGAAAGAC | GTCCC | S | S | S | S | S | S | S | S |
| IM216A | C**G**CAC | CA**T**GTC | G**T**AAAG**T**C | GT**T**CC | **>512** | **>1024** | **>256** | 64 | **>8** | **>64** | 16 | 16 |
| IM914 | CACAC | CAAGTC | GGAAAGAC | GTCCC | 4 | **128** | 32 | 4 | 0.12 | 4 | | |
| IM105 | CACAC | CAAGTC | GGAAAGAC | GTCCC | 4 | **128** | 32 | 4 | 0.12 | 4 | 1 | 2 |
| | * *** | ** *** | * **** * | ** ** | | | | | | | | |

**C.**

| | *16S rRNA* | *23S rRNA* | | MIC (µg/ml) | |
|---|---|---|---|---|---|
| | 1054 | 2058 | TET | ERY | CLI |
| CFT073 | C**A**TGG | GG**A**AAGAC | S | S | S |
| IM239 | C**A**TGG | GG**A**AAGAC | 2 | 0.12 | 0.5 |
| IM1429 | C**A**TGG | GG**G**AAGAC | 1 | **>8** | **>16** |
| IM1389 | C**A**TGG | GG**A**AAGAC | 1 | 0.12 | 1 |
| IM1459 | **T**ATGG | GG**A**AAGAC | **>128** | 0.5 | 1 |
| IM528 | C**A**TGG | GG**A**AAGAC | 1 | 0.5 | 1 |
| IM561 | **T**ATGG | GG**A**AAGAC | 64 | 0.5 | 1 |
| IM1338 | **T**ATGG | GG**A**AAGAC | **>64** | 1 | 1 |
| | **** | ** ***** | | | |

**D.**

| | Walker A1 | Walker B1 | Walker B2 | MIC (µg/ml) | mutation |
|---|---|---|---|---|---|
| *E. italicus* IM1319 | GL**I**GRNG**R**GKTT | **Y**PLID | **LF**VWD | 2 | F137Y, Y499F, I500V |
| *E. italicus* IM1314 | GL**I**GRNG**R**GKTT | **Y**PLID | **LF**VWD | 0.5 | F137Y, Y499F, I500V |
| *E. malodoratus* IM1302 | GL**V**GRNG**R**GKTT | **F**PLID | L**Y**IWD | 2 | I37V, T500I |
| *E. faecalis* IM1438 | GL**I**GRNG**R**GKTT | **F**PLID | L**YI**WD | **>16** | T500I |
| *E. faecalis* IM1315 | GL**I**GRNG**R**GKTT | **F**PLID | L**YI**WD | **>16** | T500I |
| *E. faecalis* IM1437 | GL**I**GRNG**R**GKTT | **F**PLID | L**YI**WD | **>16** | T500I |
| *E. faecalis* V583 | GL**I**GRNG**R**GKTT | **F**PLID | L**YI**WD | **32-48** | T500I |
| *E. faecalis* IM1312 | GL**I**GRNG**H**GKTT | **F**PLID | L**YI**WD | 2 | R42H, T500I |
| *E. faecalis* IM1301 | GL**I**GRNG**R**GKTT | **F**PLID | L**YI**WD | **>16** | T500I |
| *E. faecalis* IM1305 | GL**I**GRNG**R**GKTT | **F**PLID | L**YI**WD | **>16** | T500I |
| *E. faecium* IM215A | GL**I**GRNG**R**GKTT | **F**PLID | L**YI**WD | **>16** | T500I |
| *E. faecium* HM1070 | GL**I**GRNG**R**GKTT | **F**PLID | L**YT**WD | 0.12 | |
| *E. faecium* IM1313 | GL**I**GRNG**R**GKTT | **F**PLID | L**YI**WD | **8** | T500I |
| *E. faecium* IM1478 | GL**I**GRNG**R**GKTT | **F**PLID | L**YI**WD | 16 | T500I |
| *E. faecium* IM578 | GL**I**GRNG**R**GKTT | **F**PLID | L**YI**WD | 4 | T500I |
| *E. faecium* IM1439 | GL**I**GRNG**R**GKTT | **F**PLID | L**YI**WD | 16 | T500I |
| *E. durans* IM1426 | ------------ | **F**PLID | L**YI**WD | 0.25 | Premature stop codon |
| *E. hirae* IM1454 | GL**I**GRNG**R**GKTT | **F**PLID | L**YI**WD | 16 | T500I |
| | :**** | *: ** | | | |

**E.**

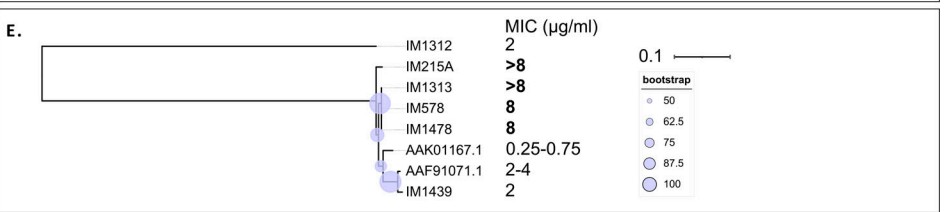

| | MIC (µg/ml) |
|---|---|
| IM1312 | 2 |
| IM215A | **>8** |
| IM1313 | **>8** |
| IM578 | 8 |
| IM1478 | 8 |
| AAK01167.1 | 0.25-0.75 |
| AAF91071.1 | 2-4 |
| IM1439 | 2 |

bootstrap: 50, 62.5, 75, 87.5, 100

**Figure 3. Polymorphisms in S12, Lsa(A), and MsrC in *16S* and *23S rRNA*.**

Shown is a section of the sequence alignment in which the mutations presumably associated with resistance are highlighted in red. **(A)** Substitution of amino acid K43 in S12 was associated with streptomycin resistance. Polymorphisms in *16S rRNA* and *23S rRNA* in strains of **(B)** *Lactobacillus paragasseri* and **(C)** *Lacticaseibacillus rhamnosus* confer resistance to different groups of antimicrobials. K-12 and CFT073 represent *Escherichia coli* strains. **(D)** Polymorphisms in key motifs of Lsa(A) and homologs were associated with clindamycin resistance (SNPs highlighted in red) and susceptibility (SNPs highlighted in purple). **(E)** Phylogenetic tree of MsrC protein sequences of *E. faecium* strains. Shown are the minimum inhibitory concentrations (MICs) of erythromycin. The tree was rooted with an outgroup (*E. faecalis* IM1312). The exceeded cut-off MICs are shown in bold. MIC, minimum inhibitory concentration; S, susceptible; GEN, gentamicin; KAN, kanamycin; NEO, neomycin; TET, tetracycline; ERY, erythromycin; CHL, chloramphenicol; AMP, ampicillin; QDA, quinupristin/dalfopristin; LIN, linezolid.

chloramphenicol, quinupristin/dalfopristin, and linezolid. The coverage and variances of *16S* and *23S rRNA* SNPs have been validated by mapping the sequenced reads to the assembled sequences (Table S3).

As reported before, enterococci exhibit clindamycin resistance because of the T500I substitution in the Walker B2 motif of the intrinsic protein Lsa(A) or its homologs (18). The susceptibility of our strains was likely related to novel mutations in key motifs of these

**Table 1. Amino acids of PBP5 proteins associated with ampicillin susceptibility of *E. faecium*.**

| Strains | Amino acid | | | | | | | | | | | | | | | | | | | | | S | R | MIC (μg/ml) |
|---|---|---|---|---|---|---|---|---|---|---|---|---|---|---|---|---|---|---|---|---|---|---|---|---|
| | 24 | 27 | 34 | 66 | 68 | 85 | 100 | 144 | 172 | 177 | 204 | 216 | 324 | 466 | 485 | 496 | 499 | 525 | 586 | 629 | 667 | | | |
| Com15[a] | V | S | R | G | A | E | E | K | T | L | D | A | T | / | M | N | A | E | V | E | P | 21 | 0 | 0.5–1 |
| IM1313 | V | S | R | G | A | E | E | K | T | L | D | A | T | / | M | N | A | E | V | E | P | 21 | 0 | 1 |
| IM1478 | V | S | R | G | A | E | E | K | T | L | D | A | T | / | M | N | A | E | V | E | P | 21 | 0 | 2 |
| IM578 | V | S | R | G | A | E | E | K | T | L | D | A | T | / | M | N | A | E | V | E | P | 21 | 0 | 1 |
| IM215A[b] | A | G | R | G | A | E | E | Q | A | L | D | A | A | / | M | N | A | E | V | E | P | 16 | 5 | >16 |
| IM1439 | A | G | Q | E | A | E | Q | Q | A | I | D | S | A | / | M | K | I | D | V | E | P | 8 | 13 | 2 |
| TX2043 | A | G | Q | E | A | E | Q | Q | A | I | D | S | A | / | M | K | T | D | V | E | P | 8 | 13 | 4 |
| C68[c] | A | G | Q | E | T | D | Q | Q | A | I | G | S | A | S | A | K | T | D | V | V | S | 1 | 20 | 256 |

The WT amino acids are shaded green, and amino acid changes, red. MIC, minimum inhibitory concentration; S, susceptible; R, resistant.
[a]PBP-S.
[b]Insertion upstream of *PBP5* (transposase).
[c]PBP-R.

**Table 2. Known or novel mutations associated with resistance to antimicrobials not included in the EFSA list ([7]).**

| Protein | Amino acid change | Phenotype | Species of origin |
|---|---|---|---|
| DfrG | F98Y/L | trimethoprim | *Staphylococcus aureus* ([22]) |
| DfrG | P21A | trimethoprim | *E. coli* ([23]) |
| DfrG | N/H23Y, D27E, A7S (*E. coli* numbering) | trimethoprim | novel mutations in the active site |
| GyrA | S83T | ciprofloxacin | *E. coli* ([24]) |
| FusA | V90I, G451A/S, H457Q, L461I/M | fusidic acid | *Staphylococcus aureus* ([25]) |
| FolP | V48I | sulphamethoxazole | *Mycobacterium leprae* ([26]) |
| LiaF | S48Y | daptomycin | novel mutation |
| LiaS | G226E, V351I | daptomycin | novel mutations |
| LiaR | E45V | daptomycin | novel mutation |
| GdpD | A249T, P307Q, F478L, D552N | daptomycin | novel mutations |

proteins (see Fig 3D). Furthermore, mutations in *lsa(A)* also affect the MIC of streptogramin, as observed in a non-starter isolate *E. faecalis* IM1301 that carried a previously reported substitution in the −10 promoter region (A-131T) ([19]).

The phylogenetic tree of intrinsic MsrC proteins (Fig 3E and Table S2) indicates that variations in sequence (including a novel substitution in a Walker A1 motif, T45S) may have an impact on erythromycin MIC and resistance in *E. faecium*. A WT strain (TX1330, AAK01167.1) had a MIC of 0.25–0.75 μg/ml, whereas our strains coding for mutated MsrC exhibited MICs of at least 2 μg/ml. Resistant strains (MIC ≥ 8 μg/ml) contained additional mutations that were reflected in two clades of the phylogenetic tree, the erythromycin-susceptible and erythromycin-resistant strains.

*E. faecium* strains exhibited ampicillin or penicillin resistance as a result of variations in 20 (or 21) amino acids of PBP5 or its promoter region described before ([20], [21]). Despite a hybrid PBP5 sequence (see Table 1), the probiotic strain *E. faecium* IM215A exhibited higher ampicillin and penicillin MICs compared with other *E. faecium* strains. We hypothesise that an insertion, which bears partial similarity to a transposase gene, between regions −10 and −35 of the *PBP5* promoter affects overexpression of the gene and leads to resistance in this strain.

Resistance to antimicrobials not included in the EFSA list ([7]) was commonly associated with known or novel mutations (Table 2). Interestingly, most species of LAB do not carry a FolP homolog, which we believe to be a reason for the extreme MICs of sulphamethoxazole.

### Phenotype–genotype agreement

In total, 1,496 phenotypic tests were performed for 103 strains, yet resistance and susceptibility could be determined for 1,314 MICs. The resulting catalogue is shown schematically in Fig 4 and described in detail in Table S2. We observed an overall high agreement (92.4%) between the presence and absence of (candidate) ARGs and mutations and the corresponding phenotypic resistance or susceptibility, respectively (Table 3). Phenotypic resistance was validated in 65.0% of the cases by genetic analyses. In fact, all exceeded cut-off values for six antibiotics could be elucidated (Table 3). All but three acquired resistance genes (*tetW*) are expressed in phenotypic resistance.

All in all, our method for predicting phenotype from genotypic data was only partially efficient. Even though positive (97.8%) and

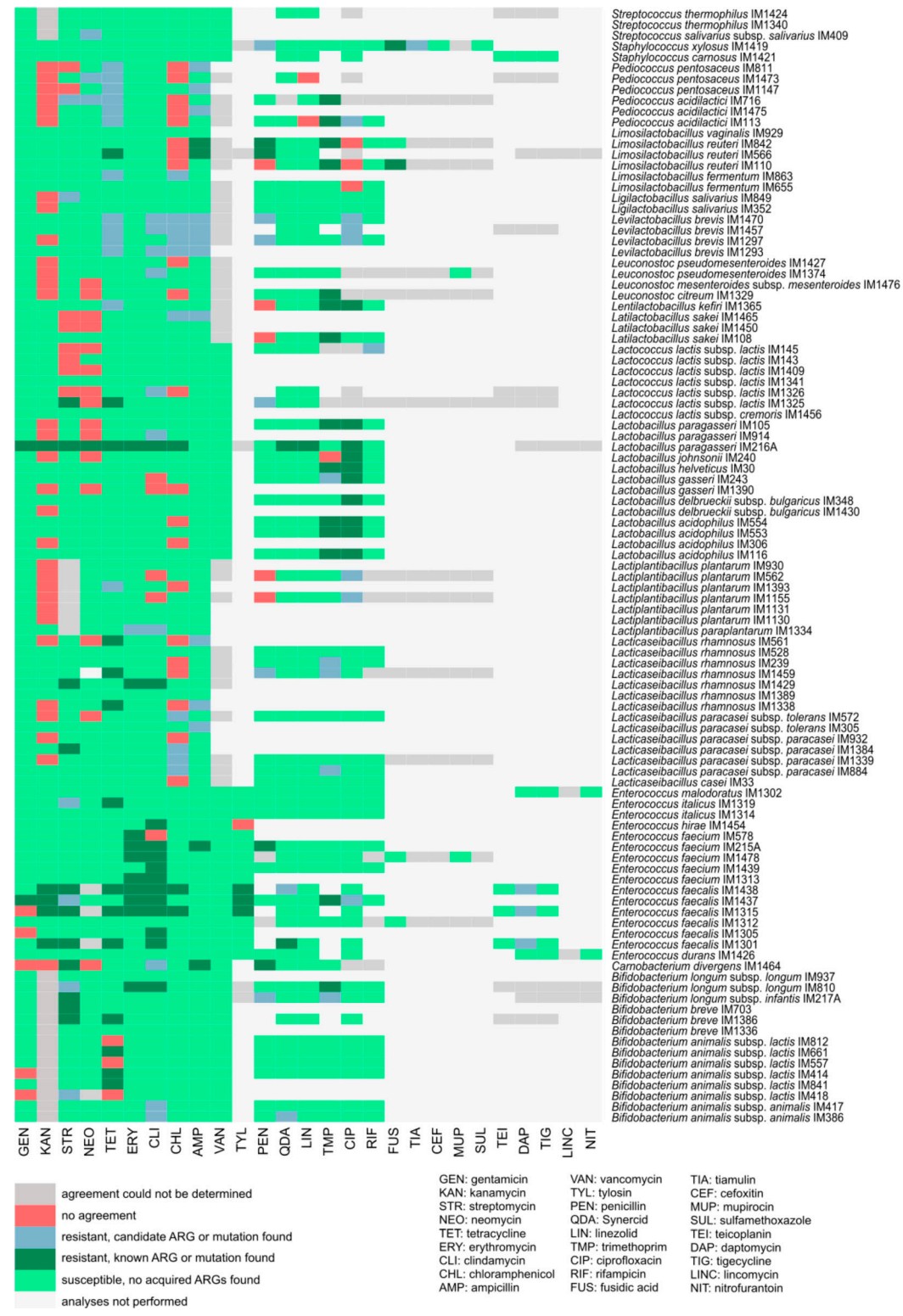

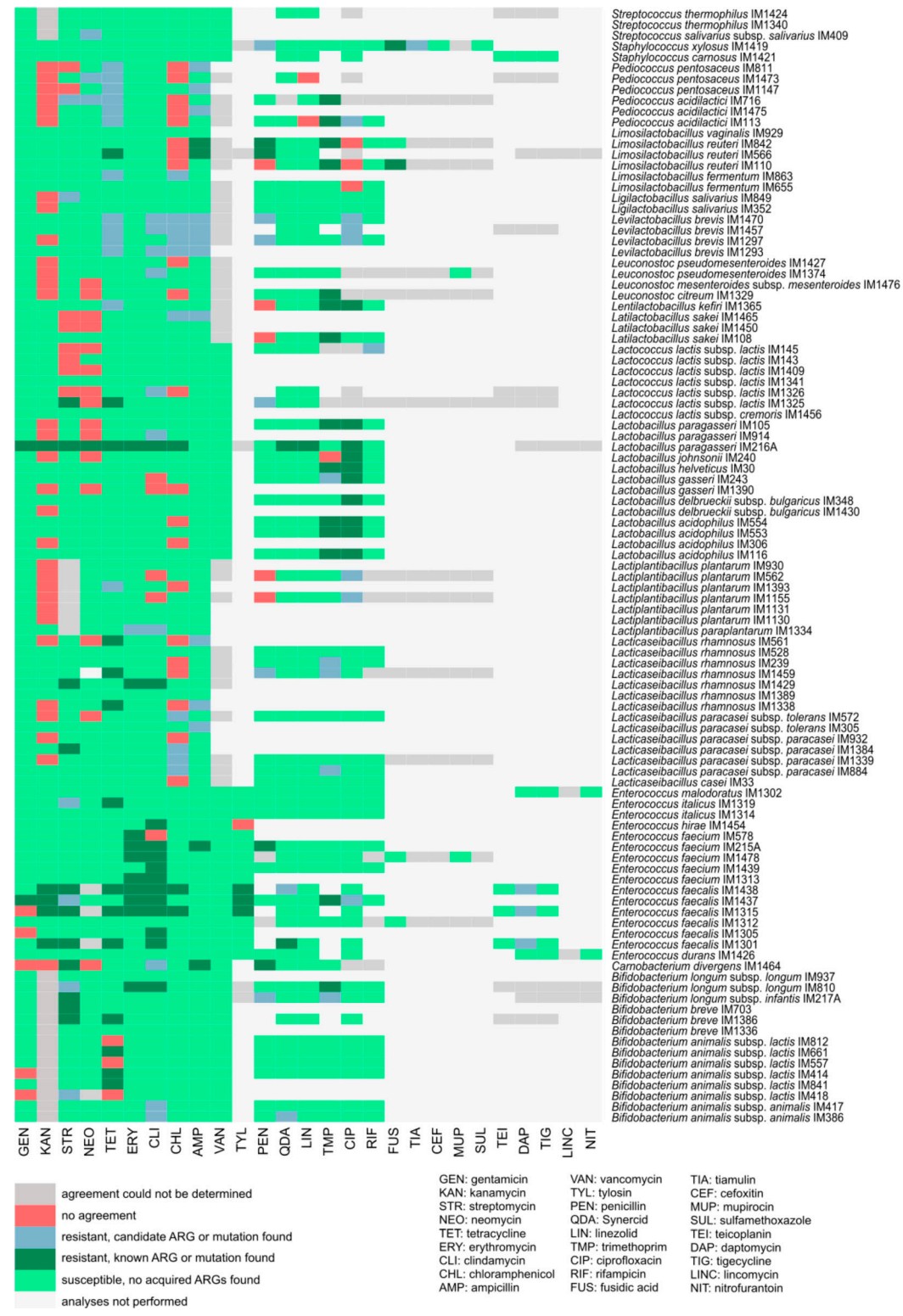 shows a heatmap with the following strain labels (rows, top to bottom):

*Streptococcus thermophilus* IM1424
*Streptococcus thermophilus* IM1340
*Streptococcus salivarius* subsp. *salivarius* IM409
*Staphylococcus xylosus* IM1419
*Staphylococcus carnosus* IM1421
*Pediococcus pentosaceus* IM811
*Pediococcus pentosaceus* IM1473
*Pediococcus pentosaceus* IM1147
*Pediococcus acidilactici* IM716
*Pediococcus acidilactici* IM1475
*Pediococcus acidilactici* IM113
*Limosilactobacillus vaginalis* IM929
*Limosilactobacillus reuteri* IM842
*Limosilactobacillus reuteri* IM566
*Limosilactobacillus reuteri* IM110
*Limosilactobacillus fermentum* IM863
*Limosilactobacillus fermentum* IM655
*Ligilactobacillus salivarius* IM849
*Ligilactobacillus salivarius* IM352
*Levilactobacillus brevis* IM1470
*Levilactobacillus brevis* IM1457
*Levilactobacillus brevis* IM1297
*Levilactobacillus brevis* IM1293
*Leuconostoc pseudomesenteroides* IM1427
*Leuconostoc pseudomesenteroides* IM1374
*Leuconostoc mesenteroides* subsp. *mesenteroides* IM1476
*Leuconostoc citreum* IM1329
*Lentilactobacillus kefiri* IM1365
*Latilactobacillus sakei* IM1465
*Latilactobacillus sakei* IM1450
*Latilactobacillus sakei* IM108
*Lactococcus lactis* subsp. *lactis* IM145
*Lactococcus lactis* subsp. *lactis* IM143
*Lactococcus lactis* subsp. *lactis* IM1409
*Lactococcus lactis* subsp. *lactis* IM1341
*Lactococcus lactis* subsp. *lactis* IM1326
*Lactococcus lactis* subsp. *lactis* IM1325
*Lactococcus lactis* subsp. *cremoris* IM1456
*Lactobacillus paragasseri* IM105
*Lactobacillus paragasseri* IM914
*Lactobacillus paragasseri* IM216A
*Lactobacillus johnsonii* IM240
*Lactobacillus helveticus* IM30
*Lactobacillus gasseri* IM243
*Lactobacillus gasseri* IM1390
*Lactobacillus delbrueckii* subsp. *bulgaricus* IM348
*Lactobacillus delbrueckii* subsp. *bulgaricus* IM1430
*Lactobacillus acidophilus* IM554
*Lactobacillus acidophilus* IM553
*Lactobacillus acidophilus* IM306
*Lactobacillus acidophilus* IM116
*Lactiplantibacillus plantarum* IM930
*Lactiplantibacillus plantarum* IM562
*Lactiplantibacillus plantarum* IM1393
*Lactiplantibacillus plantarum* IM1155
*Lactiplantibacillus plantarum* IM1131
*Lactiplantibacillus plantarum* IM1130
*Lactiplantibacillus paraplantarum* IM1334
*Lacticaseibacillus rhamnosus* IM561
*Lacticaseibacillus rhamnosus* IM528
*Lacticaseibacillus rhamnosus* IM239
*Lacticaseibacillus rhamnosus* IM1459
*Lacticaseibacillus rhamnosus* IM1429
*Lacticaseibacillus rhamnosus* IM1389
*Lacticaseibacillus rhamnosus* IM1338
*Lacticaseibacillus paracasei* subsp. *tolerans* IM572
*Lacticaseibacillus paracasei* subsp. *tolerans* IM305
*Lacticaseibacillus paracasei* subsp. *paracasei* IM932
*Lacticaseibacillus paracasei* subsp. *paracasei* IM1384
*Lacticaseibacillus paracasei* subsp. *paracasei* IM1339
*Lacticaseibacillus paracasei* subsp. *paracasei* IM884
*Lacticaseibacillus casei* IM33
*Enterococcus malodoratus* IM1302
*Enterococcus italicus* IM1319
*Enterococcus italicus* IM1314
*Enterococcus hirae* IM1454
*Enterococcus faecium* IM578
*Enterococcus faecium* IM215A
*Enterococcus faecium* IM1478
*Enterococcus faecium* IM1439
*Enterococcus faecium* IM1313
*Enterococcus faecalis* IM1438
*Enterococcus faecalis* IM1437
*Enterococcus faecalis* IM1315
*Enterococcus faecalis* IM1312
*Enterococcus faecalis* IM1305
*Enterococcus faecalis* IM1301
*Enterococcus durans* IM1426
*Carnobacterium divergens* IM1464
*Bifidobacterium longum* subsp. *longum* IM937
*Bifidobacterium longum* subsp. *longum* IM810
*Bifidobacterium longum* subsp. *infantis* IM217A
*Bifidobacterium breve* IM703
*Bifidobacterium breve* IM1386
*Bifidobacterium breve* IM1336
*Bifidobacterium animalis* subsp. *lactis* IM812
*Bifidobacterium animalis* subsp. *lactis* IM661
*Bifidobacterium animalis* subsp. *lactis* IM557
*Bifidobacterium animalis* subsp. *lactis* IM414
*Bifidobacterium animalis* subsp. *lactis* IM841
*Bifidobacterium animalis* subsp. *lactis* IM418
*Bifidobacterium animalis* subsp. *animalis* IM417
*Bifidobacterium animalis* subsp. *animalis* IM386

Column labels (left to right): GEN, KAN, STR, NEO, TET, ERY, CLI, CHL, AMP, VAN, TYL, PEN, QDA, LIN, TMP, CIP, RIF, FUS, TIA, CEF, MUP, SUL, TEI, DAP, TIG, LINC, NIT

Legend:
- agreement could not be determined
- no agreement
- resistant, candidate ARG or mutation found
- resistant, known ARG or mutation found
- susceptible, no acquired ARGs found
- analyses not performed

GEN: gentamicin
KAN: kanamycin
STR: streptomycin
NEO: neomycin
TET: tetracycline
ERY: erythromycin
CLI: clindamycin
CHL: chloramphenicol
AMP: ampicillin

VAN: vancomycin
TYL: tylosin
PEN: penicillin
QDA: Synercid
LIN: linezolid
TMP: trimethoprim
CIP: ciprofloxacin
RIF: rifampicin
FUS: fusidic acid

TIA: tiamulin
CEF: cefoxitin
MUP: mupirocin
SUL: sulfamethoxazole
TEI: teicoplanin
DAP: daptomycin
TIG: tigecycline
LINC: lincomycin
NIT: nitrofurantoin

**Figure 4.  Phenotype–genotype agreement analysis of 103 strains of lactic acid bacteria and bifidobacteria.**
In cases where no cut-off minimum inhibitory concentration was defined and in cases where the minimum inhibitory concentration was outside the concentration range of the microdilution test, agreement was not determined (shown in dark grey). ARG, resistance gene; Synercid, quinupristin/dalfopristin.

**Table 3. Phenotype–genotype agreement analysis of 103 strains for individual antibiotics.**

| | Phenotype–genotype agreement (%)[a] | Validated phenotypic resistance (%)[b] |
|---|---|---|
| Gentamicin | 95.1 | 28.6 |
| Kanamycin | 64.0 | 13.9 |
| Streptomycin | 91.7 | 68.0 |
| Neomycin | 84.7 | 21.1 |
| Tetracycline | 97.1 | 100 |
| Erythromycin | 100 | 100 |
| Clindamycin | 95.1 | 85.2 |
| Chloramphenicol | 79.6 | 38.2 |
| Ampicillin | 100 | 100 |
| Vancomycin | 100 | / |
| Tylosin | 93.8 | 75.0 |
| Penicillin | 90.2 | 66.7 |
| Quinupristin/dalfopristin | 100 | 100 |
| Linezolid | 96.8 | 33.3 |
| Trimethoprim | 98.0 | 95.0 |
| Ciprofloxacin | 94.2 | 85.0 |
| Rifampicin | 100 | 100 |
| Total | 92.4 | 65.0 |

[a]Genotype and phenotype matched when the susceptible or resistant phenotype reflected the absence or presence of (candidate) ARG(s) or mutation(s), respectively.
[b]The proportion of exceeded minimum inhibitory concentrations (phenotypic resistance) validated by genetic analyses.

negative (91.3%) predictive values and specificity (99.6%) were high, sensitivity was lower (64.3%).

### Genetic environment of the ARGs

The mobility of ARGs was estimated with the aid of the MGE analysis. Importantly, genomic island, a region of foreign origin indicative of horizontal gene transfer (27), was found in the genetic environment of acquired ARGs and one candidate ARG (Fig S1). In general, intrinsic ARGs were devoid of MGEs. This implies that the risk of horizontal transmission of intrinsic antimicrobial resistance can be considered minimal.

Genetic organisation of the discovered MGEs is depicted in Fig 5. Analysis demonstrated that enterococci frequently carry MGEs. For example, *tet*(M) is encoded on an integrative and conjugative element, Tn*916* (24.6 kbp) (Fig 5A), whereas *tet*(L), which is associated with the mobility genes *pre*/*mob* and *repB*, resides on an incomplete element that shows sequence similarity to a segment of Tn*6079* (Fig 5B). *ANT*(6)-*Ia*, SAT-4, *APH*(3')-*IIIa*, *cat*, and *erm*(B) genes are located on an element similar to the enterococcal plasmid pRE25 (Fig 5C). The full-length plasmid was not recovered. Moreover, we discovered that *dfrG* is associated with a MGE that showed strong homology to a short segment of ICESauTW20-2 from *Staphylococcus aureus* and *tet*(U) with a putative novel plasmid (Fig 5D).

Our data show that small genomic islands can be found in bifidobacterial genomes, including in probiotics *B. animalis* subsp. *lactis* (Fig 5E). A novel genomic island containing *erm*(49) and three

genes of unknown function was discovered in *B. longum* strain from a dietary supplement (Fig 5F), whereas the probiotic *B. breve* strain carried a genomic island consisting of three coding sequences (*tnpV*, *tet*(O), and the *RNA polymerase sigma factor*) (Fig 5G) and is also present in ICESsuLP081102 from *Streptococcus suis*.

The probiotic *Limosilactobacillus reuteri* strain has a tetracycline (pLR581, 12.2 kbp; Fig 5H) and a lincomycin (pLR585, 14.2 kbp; Fig 5I) resistance plasmid typical of the widely used probiotic strain *Limosilactobacillus reuteri* SD2112 (28). Similarly, *tet*(S) and *ANT*(6)-*Ia* reside on a 49,741-bp contig that is a putative plasmid (carries *repA*) (Fig 5J). Interestingly, the results indicate that *ANT*(6)-*Ia* is actually located within a novel integrative and mobilisable element (harbours an integrase and a relaxase, but lacks type IV secretion system genes) with partial homology to ICESsu(SC84) from *S. suis*. The integrative and mobilisable element is delimited by the potential *attL* (2098968..2098982, cgattttttgattttt) and *attR* (2117014..2117028) site-specific attachment sites. The genetic environment of *tet*(S) on the contrary resembles a composite transposon bounded at both ends by an insertion sequence IS1216, which is also present on plasmid pLraf_19_4S_1 in *L. raffinolactis*.

Surprisingly, only one candidate ARG (*arr-4*) was carried by a putative MGE (Fig 5K). Its genomic island consists of 26 coding sequences that share significant sequence similarity with a putative phage-inducible chromosomal island bIL310. A homologous integrase, an accessory gene, a regulatory region, and hypothetical genes were found. The element is flanked by the putative *attL* (818..835, cgcttttttactacgtt) and *attR* (18120..18137) sequences.

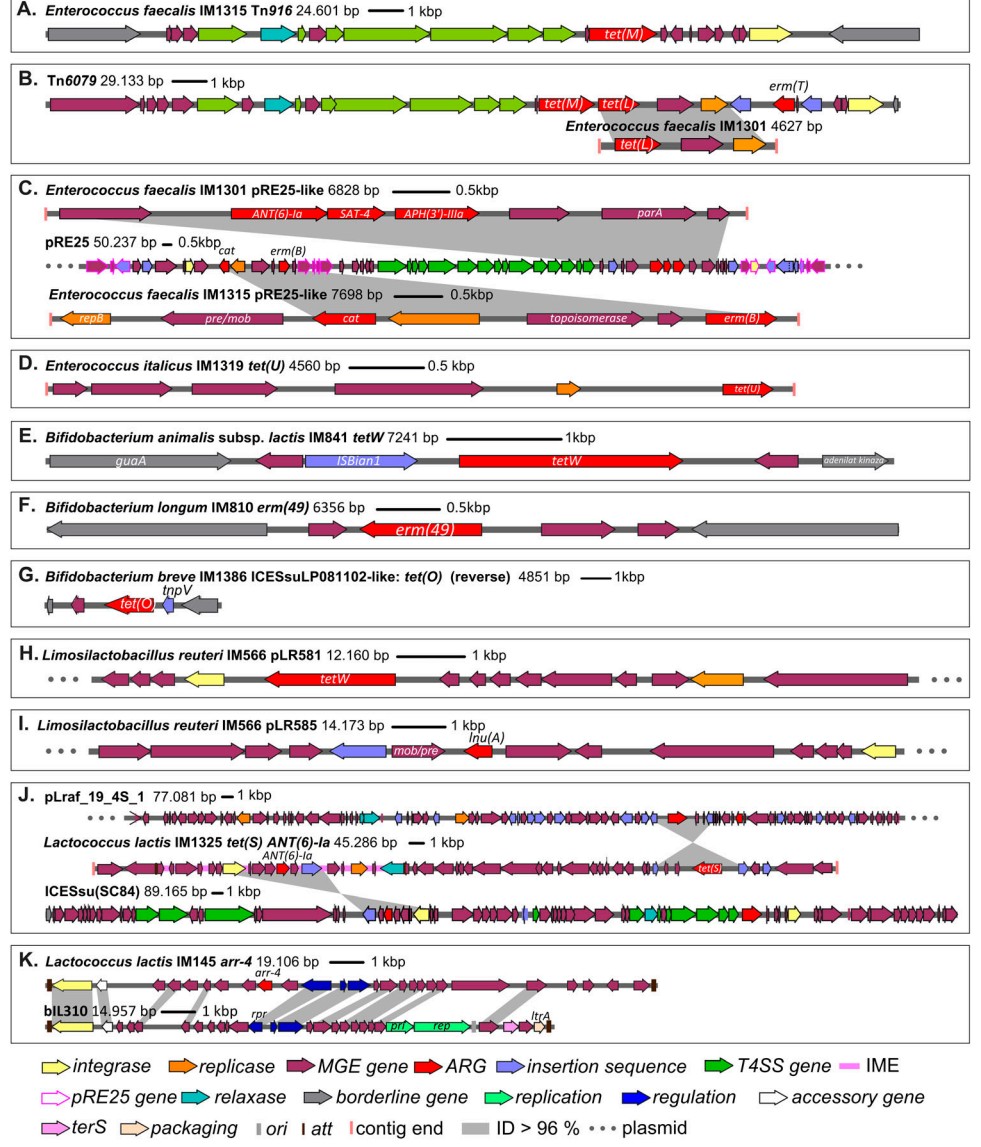

**Figure 5. Genetic organisation of the detected mobile genetic elements.**
**(A)** Gene *tet(M)* resides on Tn*916*. **(B)** Gene *tet(L)* is located on an incomplete element that shows sequence similarity to a segment of Tn*6079*. **(C)** *ANT(6)-Ia*, *SAT-4*, *APH(3′)-IIIa*, *cat*, and / or *erm(B)* are located on elements similar to the enterococcal plasmid pRE25. **(D)** Gene *tet(U)* was associated with a putative novel plasmid. Small genomic islands were found in **(E)** strains of *B. animalis* subsp. *lactis*, **(F)** *B. longum* IM810, and **(G)** *B. breve* IM1386. Probiotic bacterium *Limosilactobacillus reuteri* IM566 carries plasmids **(H)** pLR581 and **(I)** pLR585. **(J)** Genes *tet(S)* and *ANT(6)-Ia* reside on a putative plasmid. **(K)** Candidate *arr-4* is on a putative phage-inducible chromosomal island. Gene function was determined using BLAST and HMMER3, whereas genetic organisation was prepared using snapgene-viewer. ARG, antimicrobial resistance gene; ID, BLAST identity; IME, integrative and mobilisable element; T4SS, type IV secretion system.

## Discussion

Because LAB and bifidobacteria are a potential source of antibiotic resistance for gut bacteria, including pathogens, commercial strains should not carry mobile ARGs (6). However, data on resistance mechanisms, especially intrinsic and mutational resistance, are lacking. The main objective of our study was to identify the potential underlying mechanisms of the observed phenotypic resistance in 103 LAB and bifidobacteria and to assess the transferability potential using comparative genomics.

We confirmed that phenotypic resistance is a common trait in LAB and bifidobacteria, which has been described by numerous authors (2, 3, 4, 29, 30, 31). However, genetic analyses revealed that intrinsic resistance in LAB and bifidobacteria was more prevalent than acquired resistance. In accordance with the Qualified Presumption of Safety status requirement (6), acquired ARGs were not common in strains intentionally added to the agro-food chain.

Nevertheless, several probiotic bifidobacteria harboured tetracycline or erythromycin resistance genes, the presence of which on mobile elements raises the possibility of horizontal spread. The *tetW*, *tet(O)*, and *erm(49)* genes were reported in *Bifidobacterium* sp. before (32). In our recent study, we reported limited mobility of *tetW* and *erm(49)* in the metagenomic sequences of the human gut microbiota, as they were not widely disseminated and were not found outside the species of origin (33), suggesting that these two genomic islands do not pose a serious threat to food safety. The *tet(O)* genomic island, on the contrary, had a high transmission potential (33) and thus poses a risk if consumed. In accordance, Martínez et al reported the rare occurrence of *erm(49)* in the microbiomes of adults and infants (34), whereas *tet(O)* was frequently detected (35).

Our results suggest that foodborne *E. faecalis* strains play an important role in the spread of resistance. For example, *E. faecalis* strains harboured Tn*916*, which is responsible for much of the tetracycline resistance in the gut microbiota, even in pathogenic strains (36, 37), and thus poses a risk of transfer. Similarly, the pRE25 multidrug resistance plasmid was conjugated into the chromosomes of *E. faecalis*, *L. lactis*, and *Listeria innocua* (38). In concordance with our results, *tet*(*U*) was reportedly located on a small plasmid in *E. faecium* (39) and *tet*(*S*) near the transposase(s) IS1216 on a plasmid of *L. lactis*, *E. faecium*, and *S. dysgalactiae* (40). Unlike acquired ARGs, intrinsic ARGs are considered to have minimal potential for horizontal spread (12), which we confirmed by the MGE analysis.

Resistance data for many genera of LAB and bifidobacteria are not as extensive as for pathogenic bacteria; thus, fewer ARGs are available in the databases. Consequently, lower BLAST similarities are expected to be found. We uncovered numerous candidate ARGs (Fig 2 and Table S1), but their effect on the resistant phenotype needs to be verified in vitro. To the best of our knowledge, this is the first report of these genes in LAB and bifidobacteria. Surprisingly, a candidate *arr*-4 gene in *L. lactis* IM145, which we presume to encode rifampicin resistance, resides on the putative phage-inducible chromosomal island (Fig 5K). Compared with a typical prophage genome, a phage-inducible chromosomal island is smaller in size given that it does not code for capsid and lytic proteins, which we did not detect. These elements were reported in staphylococci, lactococci, pneumococci, streptococci, and enterococci and may contain genes for diverse metabolic activities or resistance genes (41).

Our study highlights that mutations of chromosomal genes (Fig 3 and Table S2) that are not considered a hazard (12) may be a frequent mechanism of resistance in LAB and bifidobacteria. The use of probiotic strains with mutational resistance may be beneficial, as they are known to help restore the natural microbiota after antibiotic therapy and reduce the severity of antibiotic-associated diarrhoea (42). Consistent with reports on fitness cost (43), polymorphisms in the active site of *16S* or *23S rRNA* were not common (Fig 3B and C). To our surprise, probiotic *Lactobacillus paragasseri* IM216A carried multiple mutations in *16S* and *23S rRNA*, which were not previously reported in this species. The A1408G mutation in *16S rRNA* causes aminoglycoside resistance (44) and does not result in a significant fitness cost compared with the lethal A1408C and A1408U mutations (43). We believe that a novel A1197T polymorphism leads to tetracycline resistance because this nucleotide is involved in hydrogen bonding with the drug (45). This strain also carried *23S rRNA* mutations (G2057T, C2610T, and A2062T) that were found in *Legionella pneumophila* (46), *Streptococcus pneumoniae* (47), and *Mycoplasma hominis* (48), respectively. Furthermore, strains of *Lacticaseibacillus rhamnosus* also had SNPs in *16S* (C1054T) or *23S rRNA* (A2058G), which is in agreement with reports for *S. pneumoniae* and *Lacticaseibacillus rhamnosus* (49, 50). The A986T SNP in *16S rRNA* has been described in the tetracycline-resistant mutant of *Mycoplasma pneumoniae* FH (51) and may therefore be linked to the tetracycline-resistant phenotypes observed in our strains. Observed S12 mutations were reported in *M. tuberculosis* (52) and *B. breve* Yakult (53), but not in lactobacilli. Similarly, the *rsmG* mutations were reported in other species (8, 54, 55), but not in LAB or bifidobacteria.

Overall, the correspondence of genotypes and phenotypes in our study was high (in 92.4%) (Fig 4 and Table 3), but further genetic studies are needed to determine the unexplained phenotypic

resistance. In accordance with our findings, Duranti et al (2017) reported a good correspondence between phenotype and genotype for type strains of bifidobacteria (2), whereas for type strains of lactobacilli, the agreement was lower (67%) (4). On the contrary, high agreement is usually reported for enterococci (9) as a result of more thoroughly characterised resistance mechanisms. Different methods for detecting ARGs and similarity cut-off values chosen, and additional screening for mutations, may also explain the observed discrepancies. Technical recommendations and requirements for whole-genome sequencing and analysis recently published by EFSA (56) are indeed an important step towards harmonisation of future studies.

In conclusion, our findings improve our understanding of the resistance mechanisms in LAB and bifidobacteria. We identified several mobile ARGs that pose a risk of transfer to pathogenic bacteria when ingested, but the prevalence of intrinsic ARGs was greater. Because intrinsic ARGs are free of MGEs, their risk of horizontal transmission can be considered minimal. We also observed that mutations may be a common mechanism of resistance. Overall, the analyses revealed high agreement between genotype and phenotype, but further genetic studies are needed to determine the unexplained phenotypic resistance. Our study presents a basis for risk assessment analyses that will ultimately ensure the safety of products used in human and animal nutrition in terms of antimicrobial resistance.

# Materials and Methods

### Bacterial strains

LAB and bifidobacteria were isolated from dietary supplements, starter and protective cultures, feed additives, human milk or colostrum, and fermented products (n = 66) or were obtained from the manufacturer or from a culture collection (n = 17). In addition, 20 probiotic and starter strains examined in our previous study (11) were reanalysed to provide additional data on candidate ARGs, mutations, and genotype–phenotype agreement. Collectively, 103 isolates were analysed (Table S2).

Serially diluted samples were cultured on the selective agar media (MRS, M17, Rogosa [Merck], and/or TOS-MUP [Yakult Honsha]) as indicated in Table S4. Strains derived from human milk or colostrum were isolated as described by Tušar et al (57) and obtained from the culture collection of the Institute of Dairy Science and Probiotics (Biotechnical faculty, University of Ljubljana) and ZIM culture collection (https://www.zim-collection.si/), which is a member of the World Federation of Culture Collections (#810). The strains were stored at −80°C, propagated under the conditions indicated in Table S4, and subcultured twice in broth medium (1% vol/vol) before all experiments.

### Isolation of genomic DNA and identification of isolates at the species level

Genomic DNA was extracted from pure overnight cultures (1 ml) using a commercial kit (ISOLATE II Genomic DNA Kit [Bioline] or

Wizard Genomic DNA Purification Kit [Promega]). Cultures were centrifuged (3 min, 12,000$g$) (Hettich), and the pellet was resuspended in 500 $\mu$l of TE buffer containing mutanolysin (25 U/ml) and lysozyme (10 mg/ml) and incubated for 2 h at 37°C. Further steps were performed according to the manufacturer's instructions.

Strains were initially identified at the species level either by PCR using species-specific primers and protocols (Table S5) or by sequencing of the 16S rDNA genes (Microsynth). Using BLAST (58), the 16S rDNA sequences were classified to species level. The taxonomic affiliation of the strains was verified by calculating the average nucleotide identity (ANI) to the WGS of a type (or selected) strain (ANI > 95% (59)) using pyani 0.2.10 (60).

## Antimicrobial susceptibility testing

MICs of the antimicrobials (see Table S2), covering almost all major classes (7), were determined by the broth microdilution method in the LSM medium (pH = 6.7) according to the standard guidelines ISO 10932 (61). We used the precoated plates VetMIC Lact-1 and Lact-2 (Statens Veterinärmedicinska Anstalt), and Sensititre AST plates EU Surveillance *Staphylococcus* EUST, EU Surveillance *Enterococcus* EUVENC, and/or NARMS Gram Positive CMV3AGPF (Thermo Fisher Scientific). In some cases, the microtitre plates for testing tylosin, vancomycin, and/or ampicillin were prepared in-house (61). After a 48-h incubation under anaerobic conditions (bifidobacteria 72 h and enterococci and staphylococci 24 h, aerobic incubation) at the temperatures listed in Table S4, the MICs were read visually as the concentration at which growth inhibition occurred. Breakpoint values were adopted from the EFSA guidelines (7) or other published guidelines (e.g., CLSI M100-ED31, EUCAST 2021) or data (see Table S6) for antibiotics not covered by EFSA. *Lacticaseibacillus paracasei* ATCC 334, *Lactiplantibacillus plantarum* ATCC 14917, *B. longum* ATCC 15707, *E. faecalis* ATCC 29212, *E. faecalis* ATCC 51299, and *L. lactis* ATCC 19435 were used as quality control strains.

## Whole-genome sequencing and assembly

The genomes of 75 bacterial strains (Table S7) were sequenced on the Illumina MiSeq platform (v3) using the Illumina TruSeq Nano library (300-bp paired-end module, Microsynth) or the Nextera XT DNA library (250-bp paired-end module, National Laboratory of Health, Environment and Food), whereas others were retrieved from public databases (GenBank accession numbers are listed in Table S2).

Quality control and trimming and filtering of raw reads were done using FastQC 0.11.9 (Babraham Bioinformatics) and TrimmomaticPE 0.39 (parameters: LEADING:3 TRAILING:3 SLIDINGWINDOW:4: 28 MINLEN:21) (62), respectively, whereas paired-end reads were merged using FLASh 1.2.11 (parameters: −min-overlap=15) (63). The resulting high-quality reads were assembled de novo using SPAdes 3.14.0 (64) with the −careful command option. To improve genome assembly, the protocol was adjusted for some strains as indicated in Table S7. QUAST 5.0.2 (65) was used to inspect the assembly statistics. Genomes were annotated using Prokka 1.14.6 (66), the level of contamination was determined using Mash Screen 2.0 (67), and plasmids were reconstructed using MOB-suite 1.4.9 (68). A total

of 2 × 66,019,708 reads were obtained. On average, 2 × 785,691 reads of 300 bp length and 2 × 1,179,891 reads of 250 bp length were retrieved per genome, giving an average genome coverage of 148× and 172×, respectively.

The whole-genome sequencing data generated in this study have been submitted to the European Molecular Biology Laboratory under the project accession PRJEB49530.

## Sequence analysis

The ARG database consisted of five publicly available databases (CARD 3.0.8 (69), ResFinder v. 2020-02-11 (70), ARG-ANNOT V6 (71), KEGG (v. November 2017) (72), and NCBI's Bacterial Antimicrobial Resistance Reference Gene Database (73)). Redundancy was removed using CD-HIT 4.7 (parameter -c 0.99) (74). In addition, the following ARG sequences were added: EfrB (accession number WP_172504673.1), bifidobacterial aminoglycoside phosphotransferases (ABE95342.1, ABE96255.1), EfmM (ADI87521.1), LmrC (WP_166668045.1), and CAD-1 (AAV65950.1).

Genome sequences were employed to query the joint ARG database with the local version of the BLAST tool (v. 2.10.0+, parameters -evalue 1e-10, -max_target_seqs 10, query coverage ≥ 60%) using a custom script. *E. faecium* DO (accession number NZ_ACIY01000000) was used as a positive control. A gene was annotated as an ARG on the basis of the best BLAST hit with a sequence similarity threshold greater than 70%. The BLAST search criterion was selected in the way to minimise the detection rate of false positives at the expense of the true positives with lower similarities based on the BLAST alignment of the ARGs database against a test dataset SwissProt (EMBL-EBI) (Table S8) as described by Hu et al (75). The ARGs discovered by BLAST were also validated with hmmsearch (HMMER3 3.1b2, parameter -E 1e-70) (76) and hidden Markov models (v. 2020-05-13) (77). The intrinsic and acquired nature of ARGs was determined with the aid of MGE prediction and pan-genome analyses. The pan- and core-genomes were computed using Roary 3.13.0 (78). In addition to the sequenced genomes, WGS were obtained from public databases and quality-checked before the analyses. QUAST was used to extract genome statistics, Mash Screen to estimate contamination, and pyani to verify taxonomic affiliation.

Genes that had a BLAST similarity threshold between 40% and 70% (BLAST data) or an E-value less than 1E-70 (HMMER data) and matched with the observed phenotype were considered as candidate ARGs. Phylogenetic analyses of these genes were conducted using RAxML-HPC v.8 (parameters: -f a -N 100 -m PROTGAMMAAUTO -p 12345 -× 12345) (79) and CompareM 0.1.1 (80) was used to calculate average amino acid identity. All-to-all BLAST results of the discovered (candidate) ARGs were filtered and clustered into groups that indicate similar functions using a custom script and mcl (v. 14-137) (The University of Utrecht). Multiple genome alignments were constructed by progressiveMauve (81), whereas protein domain analysis was performed using the Pfam database 33.1 and HMMER3 (hmmsearch, -E 1e-10). To examine mutations in proteins previously reported to be involved in resistance (n = 24), multiple sequence alignments were generated with Clustal Ω (EMBL-EBI). Subsequently, mutations were examined manually. To validate coverage and variances of SNPs in the *16S* and *23S rRNA* genes, we have

**Table 4.** Statistical parameters of genotype–phenotype agreement analysis.

| Statistical parameter | Definition |
|---|---|
| True positive | Phenotypic resistance validated by the genetic analyses (presence of antibiotic resistance gene (ARG) or mutation). |
| False positive | Phenotypic susceptibility not validated by the genetic analyses (presence of acquired ARG). |
| True negative | Phenotypic susceptibility validated by the genetic analyses (absence of acquired ARG). |
| False negative | Phenotypic resistance not validated by the genetic analyses (absence of acquired ARG or mutation). |
| Positive predictive value | TP/(TP+FP) |
| Negative predictive value | TN/(TN+FN) |
| Sensitivity | TP/(TP+FN) |
| Specificity | TN/(TN+FP) |

mapped the sequenced reads to the assembled sequences using Bowtie2 (82).

### Phenotype–genotype agreement

A total of 1,314 MICs were considered for phenotype–genotype agreement analysis (Table S2). Genotype and phenotype matched when the susceptible or resistant phenotype reflected the absence or presence of (candidate) ARG(s) or mutation(s), respectively. Sensitivity, specificity, and predictive values of phenotype prediction based on genotypic data were calculated as indicated in Table 4.

### Genetic environment of the ARGs

The genetic environment upstream and downstream (15 coding sequences) of the (candidate) ARGs extracted with SeqKit (83) was surveyed for the presence of MGEs by performing a BLAST alignment (query coverage ≥ 80%, similarity cut-off ≥ 80%) of the flanking regions with the MGE database. A custom, comprehensive, non-redundant database of MGEs (285,059 MGE genes) consisted of integrative and conjugative/mobilisable elements, transposons, insertion sequences, plasmids, integrons, prophages, and phage-inducible chromosomal islands retrieved from public databases, including those carrying ARGs in LAB and bifidobacteria. In addition, publicly available specialised databases of MGEs were included: MobilomeDB (insertion sequences, v. September 2016) (84), PlasmidFinder (v. February 2020) (85), ICEBERG 2.0 (v. May 2018) (86), PHASTER (v. August 2019) (87), and SecReT4 (v. September 2019) (88). Additional analyses were performed using tools progressiveMauve, ICEBerg 2.0, PHASTER, and hmmsearch against the Pfam database. Genetic organisation of MGEs was visualised using the snapgene-viewer 5.2.4 (SnapGene) and/or BRIG 0.95 (89).

## Data Availability

The whole-genome sequencing data from this publication have been deposited to the European Nucleotide Archive database (https://www.ebi.ac.uk/ena/browser/home) under the project accession PRJEB49530. Databases and codes are available in Figshare at https://doi.org/10.6084/m9.figshare.c.6063839.v2

## Supplementary Information

## Acknowledgements

This research was supported by the Slovenian Research Agency (Ljubljana, Slovenia) through the Young Researchers' Program (grant numbers 6316-1/2017-273 and 603-1/2017-13), the Research Project J4-1769, and the Research Program P4-0097.

### Author Contributions

V Rozman: conceptualisation, investigation, methodology, and writing—original draft.
P Mohar Lorbeg: investigation and writing—review and editing.
P Treven: visualisation and writing—review and editing.
T Accetto: software and writing—review and editing.
S Janežič: investigation and writing—review and editing.
M Rupnik: conceptualisation, funding acquisition, and writing—review and editing.
B Bogovič Matijašić: conceptualisation, funding acquisition, and writing—review and editing.

### Conflict of Interest Statement

The authors declare that they have no conflict of interest.

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
