## [Reviewer comments · Life Science Alliance]

Life Science Alliance

Genomic insights into antibiotic resistance and mobilome of lactic acid bacteria and bifidobacteria

Vita Rozman, Petra Mohar Lorbeg, Primož Treven, Tomaž Accetto, Sandra Janežič, Maja Rupnik, and Bojana Bogovič Matijašič
DOI: <https://doi.org/10.26508/lsa.202201637>

Corresponding author(s): Vita Rozman, University of Ljubljana

Review Timeline:

Submission Date:	2022-07-28
Editorial Decision:	2022-10-12
Revision Received:	2023-01-04
Editorial Decision:	2023-01-26
Revision Received:	2023-01-30
Accepted:	2023-01-30

Scientific Editor: Novella Guidi

Transaction Report:

October 12, 2022

Re: Life Science Alliance manuscript #LSA-2022-01637-T

Vita Rozman

University of Ljubljana, Biotechnical Faculty, Department of Animal Science, Institute of Dairy Science and Probiotics

Dear Dr. Rozman,

Thank you for submitting your manuscript entitled "Genomic insights into antimicrobial resistance and mobilome of lactic acid bacteria and bifidobacteria" to Life Science Alliance. The manuscript was assessed by expert reviewers, whose comments are appended to this letter. We invite you to submit a revised manuscript addressing the Reviewer comments.

Thank you for this interesting contribution to Life Science Alliance. We are looking forward to receiving your revised manuscript.

Sincerely,

B. MANUSCRIPT ORGANIZATION AND FORMATTING:

Reviewer #1 (Comments to the Authors (Required)):

In this work entitled "Genomic insights into antimicrobial resistance and mobilome of lactic acid bacteria and bifidobacteria", the authors investigated the antimicrobial resistance phenotype and genotype of 103 LAB and bifidobacteria. Specifically, they try to define intrinsic and candidate ARGs to estimate phenotype-genotype agreement, as well as predict their potential of horizontal spread through mobile genetic elements.

The manuscript is well written, and the distribution of chapters is easy to follow. Anyhow, after a first reading, the manuscript appears very long. Thus, I suggest shortening the Results and Discussion sections, especially the chapter named "Mutations associated with antimicrobial resistance" from pages 11 to 17.

Here are some comments about the manuscript.

Lines 179-180. The authors reported identifying 331 candidate ARGs represented by 33 diverse genes. However, in Table 1 are described only 15 genes. Why were 17 genes not reported?

In figure 1, where the phenotypic resistance profile of the 103 analysed strains is reported, it will be highly informative to add values of the used cut-off to define if a strain is susceptible or resistant to a specific antibiotic. Since cut-offs are different based on the different antibiotics, it would be highly informative to visualize these data together with the resistance profiles instead of Table S1.

Lines 311-317. If I have understood correctly, the authors found that the identified resistance of antibiotics (through MIC) is validated in 92% of the cases by genetic analyses. Instead, starting from the genotype data, a lower agreement level was described. Maybe these sentences can be simplified and better described to report these data. Accordingly, the description of Table 4 can be improved.

In figure 2, regarding the dark gray squares. What do you mean by "agreement could not be determined"? Please provide some information in the figure legend.

Sequencing analysis section. How the identification of SNPs in 16S and 23S genes have been managed? Since sequenced data was obtained from short Illumina reads and assembled with Spades, usually, the resulting genomes possess a single copy of the ribosomal locus. I'm trying to underline if the authors verify that the sequenced strains do not have differences in their 16S or 23S genes within the same genome across different loci. This situation may interfere with the identification of putative ARGs. Have you mapped the sequenced reads to your assembled sequences to validate coverage and variances of SNPs?

Lines 619-624. Both BLAST and HMMER were used to identify putative ARGs. Are all genes reported in this study validated with both methodologies? It can be highlighted in the text. Furthermore, they state that candidates were defined with a similarity threshold between 40 and 70%. Is this definition based on the BLAST analysis alone? How do you manage HMMER data for these candidate ARGs?

Lines 647-648. How were the sensitivity and specificity defined? The sentence is too vague. Please report how these values were calculated.

Minor comments.

Line 55. There is a typo in "gut - a hotspot".

Line 113. *Lcb. rhamnosus* is cited for the first time with its abbreviation not specified before.

Line 140. Specify the values of the sequence similarity were indicated as "lower BLAST similarities"

Line 170, what do you mean by "which 20 were diverse"? Is this assumption based on gene homology or the kind of resistance? Please specify in the main text.

In figure 5 report the complete name *B. animalis* subsp. *lactis* instead of *B. lactis*.

Line 418. Please specify what you mean by "in all sample sources".

Reviewer #2 (Comments to the Authors (Required)):

Summary: The article presents an in-depth analysis at the genome level of potential antimicrobial resistance genes, and their

risk of transmission, in lactic acid bacteria and bifidobacterial isolates with potential use in the agri-food chain. The authors base their genotypic characterization on comparative genomics, where antimicrobial resistance genes are identified based on a high sequence similarity threshold (> 70%) with published data. In addition, they propose candidates for antimicrobial resistance genes in cases of lower similarities and by manually searching for mutations in proteins previously characterized as being involved in resistance. A total of 1,314 susceptibility or resistance determinations to a panel of 27 antibiotics were used to establish the potential of genotype to phenotype agreement, which matched with genotypes in 92.4% of the cases. However, the prediction of phenotype from genotypic data was low, although it increased when considering the mutations and candidate genes found in the analysis. But, experimental validation on the potential of these genes and mutations to confer antimicrobial resistance is lacking. In general, the results are informative and warn about the potential risk of using these strains in the agri-food sector in relation to the increase in antibiotic resistance.

Major comments:

In general, the results are very descriptive, with a lot of information, that makes them difficult to read. They seem more a combination of results and discussion rather than results. The authors should consider a simplification of the text to increase readability.

Likewise, the biological significance of this work is hidden in the long results and discussion sections. I wonder whether the authors could provide with a figure/schematic where they data fits with data from human metagenomics samples.

Minor comments:

In all figures, the meaning of the grey color is not identified in the figure legend (e.g., in figure 1, does the grey color mean the susceptibility test was not performed?)

Lane 102 and lane 530, what do the authors mean with "the scope of the study". It is very confusing, on the one hand 83 strains are in the scope of the study, on the other hand 66 strains are in the scope of the study. Why aren't the 103 strains the scope of the study?

Lane 404: the authors have not identified the underlying mechanisms but potential underlying mechanisms

Referee Cross-Comments

I subscribe to all the referee's comments.

Reviewer #3 (Comments to the Authors (Required)):

Article summary:

The authors identified and investigated the antimicrobial resistances in 103 lactic acid bacteria and bifidobacteria. Commensal bacteria like LAB and bifidobacteria can carry antimicrobial resistance genes, and when ingested (e.g., as probiotics or as part of fermented food) they may transmit resistance genes into gut microbiota including to potential pathogens. In this study, the authors examined mobile genetic elements (MGEs) using whole genome sequencing of bacterial strains that are used in food fermentation or are intentionally added into the agro-food chain.

As such survey hasn't been reported before, this is an important piece of information towards preventing the spread of antibiotic resistance genes.

The major issue for the manuscript in its current format is poor writing style leading to much unclarity. The article contains a considerable detail; yet data and text presentation needs substantial improvement. The rationale behind the experimental setup/design and hypothesis are also not clearly stated.

There is much unnecessary repetition, e.g., L181-L182 contains a list of antibiotics also stated in table 1 and in the same paragraph there are 16 genes just listed, can you refer to tab1 to make it easier for readers? This pattern continues throughout the whole manuscript.

In the results and discussion section many findings and citations seem to be simply listed without proper context or further discussion. This does not mean that these findings are irrelevant, but the manuscript needs through revision. Some specific points are listed below. But please note that this list is not exhaustive as doing so will be much work for a reviewer and is beyond the scope of scientific reviewing process.

1) L34 Please explain what is enterococci when the term was used the first time.

2) L101: The result section starts very abruptly with lots of details, without any general background information. To me it reads more like methods section. Some introductory sentences could be helpful before stating the results, such as the motivation of the experiment.

Further from line 103 to 107 it has 8 numbers in the text, which is a bit overwhelming for readers.

- 3) L102 The expression of "within the scope" is a bit too general here, what specific reason makes those 83 differ from others?
- 4) You use the microdilution method to determine MIC in your study. Can you please explain the method or cite it. How exact is the determination of MIC using this method?
- 5) How did you chose/set the cut-off values shown in Table S1? Would results differ (how much) if other cut-offs were chosen?
- 6) Figure 1: Instead of showing only susceptible/resistant... it would be nice to show the MIC values in the heatmap.

How many replicates did you do for MIC testing for each strain?

The species names are abbreviated in Figure 1 but full names were shown in Figure 2, please make sure they are unified.

- 7) Table S2: Please add a column in Tab S2 that specifies which ARGs provide resistance to which antibiotic. Especially the ones termed efflux pump, it is not immediately visible for which AB types it works.
- 8) L140 Lower similarities are? Please provide more details, or a number.
- 9) L208-L219: This part is particularly confusing regarding what are results generated within this study and what data was reported before. Similar issues are found repeatedly throughout the manuscript result section.
- 10) L236ff: Unclear phrasing. Not sure what you refer to? What is the message.
- 11) L315-322: Reads like discussion, contains too many lists, numbers etc. Why not refer to tab4?
- 12) L410 "in our strain set" means? Please specify are those 103 strains or 83 strains, for example.
- 13) L419 A lot of abbreviations, such as QPS here, were only mentioned several times (<5), please use full name for them to increase clarity of the manuscript.
- 14) L417-430 What is the discussion/conclusion here after cross comparing with literature? Read more like a summary of results here.
- 15) L487-490 The purpose of this small paragraph is not very clear.

Reviewer #1 (Comments to the Authors (Required)):

In this work entitled "Genomic insights into antimicrobial resistance and mobilome of lactic acid bacteria and bifidobacteria", the authors investigated the antimicrobial resistance phenotype and genotype of 103 LAB and bifidobacteria. Specifically, they try to define intrinsic and candidate ARGs to estimate phenotype-genotype agreement, as well as predict their potential of horizontal spread through mobile genetic elements.

1. The manuscript is well written, and the distribution of chapters is easy to follow. Anyhow, after a first reading, the manuscript appears very long. Thus, I suggest shortening the Results and Discussion sections, especially the chapter named "Mutations associated with antimicrobial resistance" from pages 11 to 17.

Many thanks to the reviewer for this positive feedback, which we appreciate. As suggested we have shortened the chapter "Mutations associated with antimicrobial resistance" (see pages 8 to 11) and other parts of the Results and Discussion sections. All changes are marked using track change in word.

Here are some comments about the manuscript.

2. Lines 179-180. The authors reported identifying 331 candidate ARGs represented by 33 diverse genes. However, in Table 1 are described only 15 genes. Why were 17 genes not reported?

Table 1 included selected important candidate resistance genes. All 33 candidate genes discovered are listed in Figure 2 and Supplemental Table S1. Consistent with the reviewer

#3 comment, we decided to exclude Table 1 to avoid repetition. All Tables headings were renamed accordingly.

3. In figure 1, where the phenotypic resistance profile of the 103 analysed strains is reported, it will be highly informative to add values of the used cut-off to define if a strain is susceptible or resistant to a specific antibiotic. Since cut-offs are different based on the different antibiotics, it would be highly informative to visualize these data together with the resistance profiles instead of Table S1.

We added the used cut-off MIC values to revised Figure 1: "Phenotypic resistance profiles of 103 lactic acid bacteria and bifidobacteria. The minimum inhibitory concentrations (MICs) are shown as a heatmap. The numbers in the heatmap represent the cut-off MICs that define whether a strain is susceptible or resistant to a particular antibiotic. Resistance is shown in bold. The names of the strains resistant to five different classes of clinically important antimicrobials are highlighted in red. FD, feed additive; HM, isolate from human milk or colostrum; NA, cut-off MIC not determined; NS, isolate of natural microbiota from fermented products (non-starter strain); P, probiotic strain; PC, protective culture; Synercid, quinupristin/dalfopristin; S, starter culture."

4. Lines 311-317. If I have understood correctly, the authors found that the identified resistance of antibiotics (through MIC) is validated in 92% of the cases by genetic analyses. Instead, starting from the genotype data, a lower agreement level was described. Maybe these sentences can be simplified and better described to report these data. Accordingly, the description of Table 4 can be improved.

We rephrased the sentences in lines 276 to 282 as follows: "We observed an overall high agreement (92.4 %) between the presence and absence of (candidate) ARGs and mutations and the corresponding phenotypic resistance or susceptibility, respectively (Table 3). Phenotypic resistances were validated in 65.0 % of the cases by genetic analyses." We also improved the description of Table 3 (see also legend below the table).

5. In figure 2, regarding the dark gray squares. What do you mean by "agreement could not be determined"? Please provide some information in the figure legend.

We added the description "In cases where no cut-off minimum inhibitory concentration (MIC) was defined and in cases where the MIC was outside the concentration range of the microdilution test, agreement was not determined (shown in dark grey)." in the figure legend. Please see the lines 909 to 911 in the revised manuscript.

6. Sequencing analysis section. How the identification of SNPs in 16S and 23S genes have

been managed? Since sequenced data was obtained from short Illumina reads and assembled with Spades, usually, the resulting genomes possess a single copy of the ribosomal locus. I'm trying to underline if the authors verify that the sequenced strains do not have differences in their 16S or 23S genes within the same genome across different loci. This situation may interfere with the identification of putative ARGs. Have you mapped the sequenced reads to your assembled sequences to validate coverage and variances of SNPs?

SNPs were identified in the 16S and 23S rRNA sequences assembled by Spades as described in methods. As pointed out, usually only single copy of 16S or 23S rRNA genes were obtained from Illumina short reads. To validate coverage and variances of SNPs, we have mapped the sequenced reads to our assembled sequences. We have included this data in revised Supplemental Table S3 and in the Results and Methods sections (see lines 229 to 231 and 607 to 608 in the revised manuscript).

7. Lines 619-624. Both BLAST and HMMER were used to identify putative ARGs. Are all genes reported in this study validated with both methodologies? It can be highlighted in the text. Furthermore, they state that candidates were defined with a similarity threshold between 40 and 70%. Is this definition based on the BLAST analysis alone? How do you manage HMMER data for these candidate ARGs?

Yes, the ARGs were validated by BLAST and HMMER. We have rephrased the text in lines 588 to 589 in the revised manuscript as follows: "The ARGs discovered by BLAST were also validated with hmmsearch (HMMER3 3.1b2, parameter -E 1e-70) [76] and Hidden Markov Models (v. 2020-05-13) [77]."

Candidate ARGs were based on BLAST data (percent identity between 40 % and 70 %) and also on HMMER data (E-value (expectation value) lower than 1E-70, because hits with a higher E-value contained many false positives). We have rephrased the text in lines 595 to 596 in the revised manuscript as follows: "Genes that had a BLAST similarity threshold between 40 % and 70 % (BLAST data) or an E-value less than 1E-70 (HMMER data) and matched with the observed phenotype were considered as candidate ARGs."

8. Lines 647-648. How were the sensitivity and specificity defined? The sentence is too vague. Please report how these values were calculated.

We have added the definition of these statistical parameters in revised Table 4. We also rephrased the sentences in lines 614 to 615 as follows: "Sensitivity, specificity, and predictive values of phenotype prediction based on genotypic data were calculated as indicated in Table 4."

Minor comments.

9. Line 55. There is a typo in "gut - a hotspot".

The typo was corrected.

10. Line 113. *Lcb. rhamnosus* is cited for the first time with its abbreviation not specified before.

This abbreviation was omitted.

11. Line 140. Specify the values of the sequence similarity were indicated as "lower BLAST similarities"

We rephrased the sentence in lines 892 to 893 as follows: "Candidate (homologous) ARGs were identified based on additional analyses of the hits with lower BLAST similarities (sequence similarity threshold between 40 % and 70 %)."

12. Line 170, what do you mean by "which 20 were diverse"? Is this assumption based on gene homology or the kind of resistance? Please specify in the main text.

We have rewritten this text to: "which 20 were diverse based on gene homology" (lines 160 to 161 in the revised manuscript).

13. In figure 5 report the complete name *B. animalis* subsp. *lactis* instead of *B. lactis*.

As suggested we reported the complete name of *Bifidobacterium animalis* subsp. *lactis*.

14. Line 418. Please specify what you mean by "in all sample sources".

This paragraph was deleted.

Reviewer #2 (Comments to the Authors (Required)):

Summary: The article presents an in-depth analysis at the genome level of potential antimicrobial resistance genes, and their risk of transmission, in lactic acid bacteria and bifidobacterial isolates with potential use in the agri-food chain. The authors base their genotypic characterization on comparative genomics, where antimicrobial resistance genes are identified based on a high sequence similarity threshold (> 70%) with

published data. In addition, they propose candidates for antimicrobial resistance genes in cases of lower similarities and by manually searching for mutations in proteins previously characterized as being involved in resistance. A total of 1,314 susceptibility or resistance determinations to a panel of 27 antibiotics were used to establish the potential of genotype to phenotype agreement, which matched with genotypes in 92.4% of the cases. However, the prediction of phenotype from genotypic data was low, although it increased when considering the mutations and candidate genes found in the analysis. But, experimental validation on the potential of these genes and mutations to confer antimicrobial resistance is lacking. In general, the results are informative and warn about the potential risk of using these strains in the agri-food sector in relation to the increase in antibiotic resistance.

We thank the reviewer for the positive feedback and comments that helped us improve our manuscript.

Major comments:

1. In general, the results are very descriptive, with a lot of information, that makes them difficult to read. They seem more a combination of results and discussion rather than results. The authors should consider a simplification of the text to increase readability. Likewise, the biological significance of this work is hidden in the long results and discussion sections. I wonder whether the authors could provide with a figure/schematic where they data fits with data from human metagenomics samples.

As suggested, we have shortened and simplified the Results and Discussion sections to improve readability and point out biological significance. The biological significance is also described in the first paragraph of the Results and Discussion sections. All changes are marked using track change in word.

The results of screening human metagenomic samples are described in detail in our recent publication in Gut microbes. We have included the following reference in the manuscript:

33. Rozman V, Mohar Lorbeg P, Treven P, et al (2022) Lactic acid bacteria and bifidobacteria deliberately introduced into the agro-food chain do not significantly increase the antimicrobial resistance gene pool. *Gut Microbes* 14: 2127438

The reference list has been updated accordingly.

Minor comments:

2. In all figures, the meaning of the grey color is not identified in the figure legend (e.g., in figure 1, does the grey color mean the susceptibility test was not performed?)

Yes, grey colour means that the susceptibility test was not performed. We added this description to all figure legends.

3. Lane 102 and lane 530, what do the authors mean with "the scope of the study". It is very confusing, on the one hand 83 strains are in the scope of the study, on the other hand 66 strains are in the scope of the study. Why aren't the 103 strains the scope of the study?

All 103 strains were in scope of the study. We rephrased the sentences in lines 486 to 492 as follows: "LAB and bifidobacteria were isolated from dietary supplements, starter and protective cultures, feed additives, human milk or colostrum, and fermented products (n = 66) or were obtained from the manufacturer or from a culture collection (n = 17). In addition, 20 probiotic and starter strains examined in our previous study [11] were reanalysed to provide additional data on candidate ARGs, mutations, and genotype-phenotype agreement. Collectively, 103 isolates were analysed (Supplementary Table S2)."; and in lines 105 to 107: "The minimum inhibitory concentrations (MICs) of up to 27 antimicrobials were tested using the broth microdilution method for 103 LAB and bifidobacteria (Fig 1)."

4. Lane 404: the authors have not identified the underlying mechanisms but potential underlying mechanisms

We changed 'underlying mechanisms' to 'potential underlying mechanisms' in all cases throughout the manuscript.

Referee Cross-Comments

I subscribe to all the referee's comments.

Reviewer #3 (Comments to the Authors (Required)):

Article summary:

The authors identified and investigated the antimicrobial resistances in 103 lactic acid bacteria and bifidobacteria. Commensal bacteria like LAB and bifidobacteria can carry antimicrobial resistance genes, and when ingested (e.g., as probiotics or as part of fermented food) they may transmit resistance genes into gut microbiota including to

potential pathogens. In this study, the authors examined mobile genetic elements (MGEs) using whole genome sequencing of bacterial strains that are used in food fermentation or are intentionally added into the agro-food chain.

As such survey hasn't been reported before, this is an important piece of information towards preventing the spread of antibiotic resistance genes.

We thank the reviewer for the helpful comments, which greatly assisted us in improving the manuscript.

The major issue for the manuscript in its current format is poor writing style leading to much unclarity. The article contains a considerable detail; yet data and text presentation needs substantial improvement. The rationale behind the experimental setup/design and hypothesis are also not clearly stated.

We have shortened and simplified the Results and Discussion sections to improve clarity and point out biological significance. The rationale behind the experimental setup/design is described in the Introduction section as well as in the beginning of the Results and Discussion sections. This study is exploratory in nature so no hypotheses were formulated, only research goals that included the discovery of potential novel mechanisms of antibiotic resistance and mobile genetic elements. All changes are marked using track change in word.

There is much unnecessary repetition, e.g., L181-L182 contains a list of antibiotics also stated in table 1 and in the same paragraph there are 16 genes just listed, can you refer to tab1 to make it easier for readers? This pattern continues throughout the whole manuscript.

We thank the reviewer for this important remark. All candidate genes discovered are listed in Figure 2, while the drug class is reported in revised Supplemental Table S1. To avoid repetition, we decided to omit Table 1. We also reduced redundant information throughout the manuscript (lines 106 to 120, 173 to 175, and 279 to 287; chapters 'Acquired ARGs' (pages 6 to 7), 'Mutations associated with antimicrobial resistance' (pages 8 to 11), and 'Genetic environment of the ARGs' (pages 12 to 14) in the revised manuscript).

In the results and discussion section many findings and citations seem to be simply listed without proper context or further discussion. This does not mean that these findings are irrelevant, but the manuscript needs through revision. Some specific points are listed

below. But please note that this list is not exhaustive as doing so will be much work for a reviewer and is beyond the scope of scientific reviewing process.

We thank the reviewer for this remark. We have thoroughly revised and improved the manuscript. Citations were revised and further discussed where necessary. The citation list was updated accordingly.

1) L34 Please explain what is enterococci when the term was used the first time.

We explained the term enterococci at first appearance (abstract and text). We rephrased the sentences in lines 72 to 74 as follows: "Acquired resistance in *Enterococcus* sp. (enterococci) is widespread and considerably well described, as some strains are important nosocomial pathogens [3, 13]."

2) L101: The result section starts very abruptly with lots of details, without any general background information. To me it reads more like methods section. Some introductory sentences could be helpful before stating the results, such as the motivation of the experiment.

The results and discussion sections were shortened and simplified. Introduction was added to the beginning of the Results section: "LAB and bifidobacteria can carry mobile ARGs, and when ingested, they can facilitate the transfer of these genes to the resident microbiota in the gut and thus to potential pathogens. Commercial strains are required to be free of acquired (mobile) ARGs [6], but data on the genetic basis of phenotypic resistances in these bacteria are limited. The main objective of our study was to identify the potential underlying mechanisms of acquired and intrinsic resistances in LAB and bifidobacteria using comparative genomics." (lines 99 to 104 in the revised manuscript).

Further from line 103 to 107 it has 8 numbers in the text, which is a bit overwhelming for readers.

We rephrased the sentences in lines 107 to 114 as follows: "We observed that resistance to kanamycin and chloramphenicol were the most common clinically relevant phenotypes (Fig 1). In contrast, a lower prevalence of resistance was seen with gentamicin, erythromycin, and ampicillin, whereas atypical vancomycin resistance [7] was not detected (Fig 1)." Resistance rates for individual antibiotics were added to the revised Figure 1.

3) L102 The expression of "within the scope" is a bit too general here, what specific reason makes those 83 differ from others?

Please refer to the question number 3 from Reviewer #2.

4) You use the microdilution method to determine MIC in your study. Can you please explain the method or cite it. How exact is the determination of MIC using this method?

As described in the manuscript (lines 529 to 547) the broth microdilution method was used to determine the MICs of antimicrobials according to the standard ISO 10932 (Milk and milk products - determination of the minimal inhibitory concentration (MIC) of antibiotics applicable to bifidobacteria and non-enterococcal lactic acid bacteria (LAB)).

We rephrased the sentence in lines 529 to 532 as follows: "MICs of the antimicrobials (see Supplementary Table S2), covering almost all major classes [7], were determined by the broth microdilution method in the LSM medium (pH = 6.7) according to the standard guidelines ISO 10932 [61]."

Briefly, bacterial strains were prepared by suspending colonies from agar plates into a sterile ¼ Ringer solution (Merck) until the turbidity reached McFarland standard 1. Bacterial suspensions were then diluted 1000-fold in the lactic acid bacteria susceptibility testing medium that was prepared from 90% Iso-Sensitest (Oxoid, Basingstoke, England) and 10% MRS broth (Merck) and distributed into the wells of the plates (100 µl). The plates were incubated as described in the manuscript. As stated in this standard, MICs are read visually. If negative and positive controls are checked and approved, growth is determined visually for each antibiotic by comparing with the positive control (preferably with a viewing device – enlarging mirror). Bacterial growth is easily detected in the mirror as a pellet at the bottom of the well. Series of well where discontinuity in growth is observed are discarded. The end point is defined as the lowest antibiotic concentration at which there is no visual growth. Because the standard ISO 10932 is cited in the manuscript, detailed description of the method is not provided.

5) How did you chose/set the cut-off values shown in Table S1? Would results differ (how much) if other cut-offs were chosen?

As described in the Methods section, strains were classified as resistant or susceptible according to the epidemiological cut-off values (ECOFFs) published by EFSA (EFSA-FEEDAP. Guidance on the characterisation of microorganisms used as feed additives or as production organisms. EFSA Journal. 2018;16(3):5206. doi:10.2903/j.efsa.2018.5206). These values are selected based on testing a large number of strains and are regularly updated by EFSA. In the EFSA guidelines, cut-off MICs are available for a limited number of species and antimicrobials (only those considered clinically important). Therefore, cut-

off values for antibiotics not covered by EFSA were selected based on other available guidelines (clinical or epidemiological, e.g. CLSI, EUCAST) or published data cited in Table S6. These values affect the resistance and susceptibility profiles. The susceptibility data for these antibiotics will be useful to EFSA in setting breakpoint MICs.

We rephrased the sentence in lines 541 to 543 as follows: "Breakpoint values were adopted from the EFSA guidelines [7] or other published guidelines (e.g. CLSI M100-ED31, EUCAST 2021) or data (see Supplementary Table S6) for antibiotics not covered by EFSA."

6) Figure 1: Instead of showing only susceptible/resistant... it would be nice to show the MIC values in the heatmap.

We showed MIC values in revised Figure 1 as a heatmap. Resistance is now shown in bold. We also added the used cut-off values to define if a strain is susceptible or resistant to a specific antibiotic. Please refer to comment number 3 from reviewer #1.

How many replicates did you do for MIC testing for each strain?

MIC testing was performed according to ISO 10932 standard (Milk and milk products - determination of the minimal inhibitory concentration (MIC) of antibiotics applicable to bifidobacteria and non-enterococcal lactic acid bacteria (LAB)), so one replicate was done and all necessary controls were included. Series of wells where discontinuity in growth was observed were discarded.

The species names are abbreviated in Figure 1 but full names were shown in Figure 2, please make sure they are unified.

We unified species names in all Figures. Please see revised Figures.

7) Table S2: Please add a column in Tab S2 that specifies which ARGs provide resistance to which antibiotic. Especially the ones termed efflux pump, it is not immediately visible for which AB types it works.

A column with antibiotics was added to revised Table S1.

8) L140 Lower similarities are? Please provide more details, or a number.

We rephrased the sentence in lines 892 to 893 as follows: "Candidate (homologous) ARGs were identified based on additional analyses of the hits with lower BLAST similarities (sequence similarity threshold between 40 % and 70 %)."

9) L208-L219: This part is particularly confusing regarding what are results generated within this study and what data was reported before. Similar issues are found repeatedly throughout the manuscript result section.

All mutations were discovered in this study. Some of these were found also in other species before but mostly not in LAB or bifidobacteria, which has clearly been stated in revised Discussion section (lines 426 to 449).

We also rephrased the sentences in lines 192 to 221 as follows: "Multiple sequence alignment of the S12 proteins revealed two mutations, K43R/N/M and K88Q (*Mycobacterium tuberculosis* numbering) in commercial streptomycin resistant strains (Fig 3A). Likewise, we discovered *rsmG* point mutations (I55A, G164V, D67N, *M. tuberculosis* numbering, and G10E, R190H, *Streptomyces coelicolor* numbering, Supplementary Table S2) involved in low-level streptomycin resistance. Although LAB are generally less susceptible to aminoglycosides, three strains (*Lactobacillus acidophilus* IM116, *L. lactis* IM1456, IM1341) exhibited a hypersusceptible phenotype. Interestingly, these strains harboured single nucleotide polymorphisms (SNPs) in the F0F1 ATPase genes (Supplementary Table S2). F0F1 ATPase is reportedly involved in aminoglycoside transport into cells [17] that could be hampered by these mutations. The effects of these mutations on resistance has yet to be confirmed *in vitro*."

10) L236ff: Unclear phrasing. Not sure what you refer to? What is the message.

We rephrased this paragraph (lines 213 to 231): "Several resistant probiotic strains had SNPs in *16S rRNA* (A1408G, C1054T, A1197T, *Escherichia coli* numbering) that presumably confer resistance to aminoglycosides or tetracycline (Fig 3B and C3). We also identified a SNP (A986T, *E. coli* numbering) near the primary tetracycline binding site in the representatives of LAB (Supplementary Table S2) displaying high-end tetracycline MICs. Among four SNPs in *23S rRNA* (Fig 3B and 3C), G2057T, A2058G, and C2610T presumably encode resistance to MLS_B, whereas A2062T to tylosin, chloramphenicol, quinupristin/dalfopristin, and linezolid. The coverage and variances of *16S* and *23S rRNA* SNPs have been validated by mapping the sequenced reads to the assembled sequences (Supplementary Table S3)."

11) L315-322: Reads like discussion, contains too many lists, numbers etc. Why not refer to tab4?

We rephrased this paragraph (lines 274 to 287): "In total, 1496 phenotypic tests were performed for 103 strains, yet resistance and susceptibility could be determined for 1314 MICs. The resulting catalogue is shown schematically in Fig 4 and described in detail in Supplementary Table S2. We observed an overall high agreement (92.4 %) between the presence and absence of (candidate) ARGs and mutations and the corresponding phenotypic resistance or susceptibility, respectively (Table 3). Phenotypic resistances were validated in 65.0 % of the cases by genetic analyses. In fact, all exceeded cut-off values for six antibiotics could be elucidated (Table 3). All but three acquired resistance genes (*tetW*) expressed in phenotypic resistance.

All in all, our method for predicting phenotype from genotypic data was only partially efficient. Even though positive (97.8 %) and negative (91.3 %) predictive values and specificity (99.6 %) were high, sensitivity was lower (64.3 %). "

12) L410 "in our strain set" means? Please specify are those 103 strains or 83 strains, for example.

This paragraph was excluded.

13) L419 A lot of abbreviations, such as QPS here, were only mentioned several times (<5), please use full name for them to increase clarity of the manuscript.

Abbreviations that were mentioned less than 5 times were omitted (QPS, PICI, ICE, IME, GI, IS, and the majority of lactic acid bacteria genera).

14) L417-430 What is the discussion/conclusion here after cross comparing with literature? Read more like a summary of results here.

We rephrased the paragraph and added the discussion and conclusions (lines 370 to 387): "In accordance with the Qualified Presumption of Safety status requirement [6], acquired ARGs were not common in strains intentionally added into the agro-food chain. Nevertheless, several probiotic bifidobacteria harboured tetracycline or erythromycin resistance genes, the presence of which on mobile elements raises the possibility of horizontal spread. The *tetW*, *tet(O)*, and *erm(49)* genes were reported in *Bifidobacterium* sp. before [32]. In our recent study, we reported limited mobility of *tetW* and *erm(49)* in the metagenomic sequences of the human gut microbiota, as they were not widely disseminated and were not found outside the species of origin [33], suggesting that these two genomic islands do not pose a serious threat to food safety. The *tet(O)* genomic island, on the other hand, had a high transmission potential [33] and thus poses a risk if consumed. In accordance, Martínez et al. (2018) reported the rare occurrence of *erm(49)* in the microbiomes of adults and infants [34], whereas *tet(O)* was frequently detected [35]."

Two references were added:

32. Cao L, Chen H, Wang Q, et al (2020) Literature-based phenotype survey and *in silico* genotype investigation of antibiotic resistance in the genus *Bifidobacterium*. *Curr Microbiol* 77:4104–4113
33. Rozman V, Mohar Lorbeg P, Treven P, et al (2022) Lactic acid bacteria and bifidobacteria deliberately introduced into the agro-food chain do not significantly increase the antimicrobial resistance gene pool. *Gut Microbes* 14: 2127438

15) L487-490 The purpose of this small paragraph is not very clear.

We deleted this paragraph.

January 26, 2023

RE: Life Science Alliance Manuscript #LSA-2022-01637-TR

Dr. Vita Rozman
University of Ljubljana
Groblje 3
Domžale 1230
Slovenia

Dear Dr. Rozman,

Thank you for submitting your revised manuscript entitled "Genomic insights into antibiotic resistance and mobilome of lactic acid bacteria and bifidobacteria". We would be happy to publish your paper in Life Science Alliance pending final revisions necessary to meet our formatting guidelines.

- please address the final Reviewer 1's comment regarding the readability of Figure 1
- please add the Twitter handle of your host institute/organization as well as your own or/and one of the authors in our system
- please upload a clean manuscript without the track changes
- there is Appendix S1 uploaded separately. Please upload it as a Supplementary Material and cite it in the manuscript text

A. FINAL FILES:

B. MANUSCRIPT ORGANIZATION AND FORMATTING:

Sincerely,

Reviewer #1 (Comments to the Authors (Required)):

The authors satisfied all my concerns. I appreciated the improvement done in Figure 1, but I had problems fully understanding the color code used. Sometimes colors and MIC values do not correspond with the reported legend (MIC (ug/ml)). Furthermore, colors used for marking MIC below or above the microdilution test are too similar to those of the MIC values, and sometimes resistance within the same species is not represented in bold for all the strains. Please make some improvements to provide a readable Figure 1.

Reviewer #2 (Comments to the Authors (Required)):

The authors have positively addressed many comments raised by the reviewers. The readability of the manuscript was a major concern, and this has improved significantly, including finding a conclusion to the work presented. Figures and figures descriptions are also improved, which helps to understand the results. It is appreciated the effort to reduce the number of results explained in plain text, which are now referred to tables.

Reviewer #3 (Comments to the Authors (Required)):

I am satisfied by the revisions.

Life Science Alliance

Reviewer #1 (Comments to the Authors (Required)):

The authors satisfied all my concerns. I appreciated the improvement done in Figure 1, but I had problems fully understanding the color code used. Sometimes colors and MIC values do not correspond with the reported legend (MIC (ug/ml)). Furthermore, colors used for marking MIC below or above the microdilution test are too similar to those of the MIC values, and sometimes resistance within the same species is not represented in bold for all the strains. Please make some improvements to provide a readable Figure 1.

We agree with the reviewer on the readability of Figure 1. We double-checked the colour code, MIC values, and resistance profiles. The colours and MIC values match in all cases and resistance is now clearly marked (R in bold). We also changed the colours of the MICs below and above the microdilution test as suggested. To improve readability, cut-off MIC values are now also presented as a heatmap instead of numbers (right column for each antibiotic, marked as C). MIC values obtained by microdilution are presented in the left column for each antibiotic (marked as M). Please see the revised Figure 1.

Reviewer #2 (Comments to the Authors (Required)):

The authors have positively addressed many comments raised by the reviewers. The readability of the manuscript was a major concern, and this has improved significantly, including finding a conclusion to the work presented. Figures and figures descriptions are

also improved, which helps to understand the results. It is appreciated the effort to reduce the number of results explained in plain text, which are now referred to tables.

Reviewer #3 (Comments to the Authors (Required)):

I am satisfied by the revisions.

January 30, 2023

RE: Life Science Alliance Manuscript #LSA-2022-01637-TRR

Dr. Vita Rozman
University of Ljubljana
Groblje 3
Domžale 1230
Slovenia

Dear Dr. Rozman,

Thank you for submitting your Research Article entitled "Genomic insights into antibiotic resistance and mobilome of lactic acid bacteria and bifidobacteria". It is a pleasure to let you know that your manuscript is now accepted for publication in Life Science Alliance. Congratulations on this interesting work.

DISTRIBUTION OF MATERIALS:

Again, congratulations on a very nice paper. I hope you found the review process to be constructive and are pleased with how the manuscript was handled editorially. We look forward to future exciting submissions from your lab.

Sincerely,
